



# Use of Genetic Algorithms for Ocean Model Parameter Optimisation

Marcus Falls[1], Raffaele Bernardello[1], Miguel Castrillo[1], Mario Acosta[1], Joan Llort[1], and Martí Galí[1]

[1]Barcelona Supercomputing Center

**Correspondence:** Marcus Falls (marcus.falls@bsc.es), Martí Galí (marti.galitapias@bsc.es)

**Abstract.** When working with Earth system models, a considerable challenge that arises is the need to establish the set of parameter values that ensure the optimal model performance in terms of how they reflect real-world observed data. Given that each additional parameter under investigation increases the dimensional space of the problem by one, simple brute-force sensitivity tests can quickly become too computationally strenuous. In addition, the complexity of the model and interactions between parameters mean that testing them on an individual basis has the potential to miss key information. As such, this work argues the need of the development of a tool that can give an estimation of parameters. Specifically it proposes the use of a Biased Random Key Genetic Algorithm (BRKGA). This method is tested using the one dimensional configuration of PISCES-v2, the biogeochemical component of NEMO, a global ocean model. A test case of particulate organic carbon in the North Atlantic down to 1000 m depth is examined, using observed data obtained from autonomous biogeochemical Argo floats. In this case, two sets of tests are run, one where each of the model outputs are compared to the model outputs with default settings, and another where they are compared with 3 sets of observed data from their respective regions, which is followed by a cross reference of the results. The results of these analyses provide evidence that this approach is robust and consistent, and also that it provides indication of the sensitivity of parameters on variables of interest. Given the deviation of the optimal set of parameters from the default, further analyses using observed data in other locations are recommended to establish the validity of the results obtained.

## 1 Introduction

The field of Earth Science has garnered much interest in recent years due to anthropogenic-driven climate change, and the increasing urgency to implement policies and technologies to mitigate its effects. As a result, Earth System Models (ESMs) have become a fundamental tool to study the impact of mutating climate dynamics and global biogeochemical cycles (Eyring et al., 2016; Anav et al., 2013; Flato, 2011). Driven by the necessity of policy makers to have increasingly reliable future climate projections, ESMs are being continuously developed resulting in highly complex and computationally demanding tools. As a result, in recent decades their capability has been strongly linked to the amount of computing power and data storage capacity





available. At present, future climate projections produced by ESMs are still hampered by both technical limitations and a

lack of knowledge of important processes (Seferian et al., 2020). Particularly, the representation of the global carbon cycle, specifically ocean biogeochemistry, suffers from many uncertainties. Moreover, the drive for realistic physical processes is pushing ESMs towards higher spatial resolution making the cost of calibrating the ocean biogeochemical component of ESMs unsustainable (Galbraith et al., 2015; Kriest et al., 2020). This is true for all ESM components that are not central to the core general circulation model (e.g. atmospheric chemistry). Thus there is a vital need for novel solutions that allows to optimise

such components in a cost-effective way in order to provide critical analyses of the evolution of the climate and answer key societal questions in relation to it (Palmer, 1999, 2014).

The tool presented here can be applied to any ESM component, although this work focuses on ocean biogeochemistry because of the many unconstrained parameters that are usually needed to numerically represent this realm of the Earth System. In particular, we focus on key biogeochemical processes that control the oceans' capacity to absorb carbon dioxide from

the atmosphere and to store it in intermediate and deep water masses over decadal to millennial timescales. These processes are usually referred to as the "biological carbon pump", which we briefly introduce here. In the sunlit layer of the ocean, phytoplankton photosynthesis transforms carbon dioxide and inorganic nutrients into organic matter. This organic matter is exported to deeper layers in the form of detrital particles of various sizes and compositions as a result of numerous processes, such as physico-chemical particle aggregation, biological repackaging into fecal pellets by predators, gravitational sinking, and

physical particle injection (Boyd et al., 2019). Along their journey, these organic detritus are decomposed back to inorganic carbon and nutrients by bacteria, but also intercepted and transformed by larger organisms, eventually re-entering the food web. The interplay between these two biological pathways and particle sinking determines the amount of inorganic carbon and nutrients released at each depth and, therefore, for how long these compounds will be locked in the ocean interior before vertical mixing and large-scale circulation bring them back to the surface. Once there, they will influence primary productivity

and the exchange of carbon with the atmosphere. Given that the oceans have absorbed around 30 % of the carbon dioxide released by human activity since preindustrial times (Gruber et al., 2019), constraining uncertainties in these biogeochemical processes is crucial to predict the future evolution of the climate system. However, their representation in models is still a challenge, in particular in the mesopelagic layer that extends between the bottom of the sunlit upper ocean and 1000 m, where around 90 % of detrital matter degradation takes place (Burd et al., 2010; Giering et al., 2014).

Ocean biogeochemistry models (OBGCM) unavoidably simplify the complexity of the real world, and normally aggregate the array of diverse species into broad groups with shared functional traits, and use empirical functions to simulate the interactions and fluxes between these groups (Fasham et al., 1990). The parameterisations are often determined by studies of plankton monocultures performed in a controlled laboratory environment (Pahlow et al., 2013), or based on sparse field measurements that can hardly represent the entire range of biogeochemical conditions (Friedrichs et al., 2007; Aumont et al., 2015).

Therefore, it is likely that model parameterisations do not reflect the complexity and diversity present in our oceans. However, the alternative of increasing model complexity to better represent real-world variability can come at a cost, as it requires the addition of parameters that may further increase uncertainty (Anderson, 2005).





In the strive for achieving simple yet universally-applicable models, parameter optimisation (PO) techniques are a key tool, as they provide an objective means to find a model parameter set that produces outputs that match well with observed datasets. However, PO (often referred to as "tuning") has traditionally been a rather subjective process, in that the model developers choose the "best" parameter sets from a more or less comprehensive array of alternative model runs. Such subjective optimisation often relied on sensitivity analyses, whereby the variations in model output variables, and their skill, were quantified after perturbing one parameter at a time. Comprehensive sensitivity analyses become increasingly unfeasible and harder to interpret as higher-order parameter interactions are considered. However, given the high computing cost of 3D OBGCM simulations, subjective criteria are still widely used to optimise OBGCMs. Although streamlined optimisation procedures for 3D OBGCMs have been proposed (Kriest et al., 2020), the establishment of consensus optimisation protocols for OBGCMs currently appears as a distant goal compared, for example, to climate models (Schmidt et al., 2017). In the case of OBGCMs, a workable alternative is to perform PO using one-dimensional (1D) model configurations, which deal only with local sources and sinks and vertical fluxes along the water column. OBGCMs have historically been developed and optimised in 1D configurations (Fasham et al., 1990; Friedrichs et al., 2007; Bagniewski et al., 2011; Ayata et al., 2013) and, in fact, state-of-the-art OBGCMs like PISCES-v2 (Pelagic Interactions Scheme for Carbon and Ecosystem Studies version 2) (Aumont et al., 2015) are 1D models embedded in a circulation model that performs tracer advection and diffusion. Optimizing OBGCMs in 1D is advantageous, as it enables a thorough exploration of the parameter space at reduced computing cost. Moreover, if parameter sets are optimised against observations that span different spatial and temporal domains, PO can also assist in the task of evaluating model portability (Friedrichs et al., 2007) and parametric uncertainty.

Attempting to constrain parameters using optimisation techniques can be difficult in situations of inadequate data or computing power (Matear, 1995; Fennel et al., 2000). However in recent years this approach has become more viable within the scientific community due to improvements in the newer generations of supercomputers and novel High Performance Computing (HPC) techniques to efficiently exploit the parallelism of those supercomputers (Casanova et al., 2011; Broekema and Bal, 2012). This provides the opportunity to run multiple simulations simultaneously, opening the way to apply PO methods, including computational and artificial intelligence, to better understand and improve model accuracy. Genetic algorithms (GA) are one such example of this, they have been applied to many global search problems and, in recent years, have also started to be used in numerical modelling to avoid the limitations of today's weather and climate models. Another example is the URANIE tool (Gaudier, 2010), which uses neural networks to learn surrogate models with a reduced computational cost. What these different algorithms have in common is the fact that they are based on iterative processes traversing a search space by applying operations on the candidate solutions with the purpose of finding a global optimum. Candidate solutions are evaluated by a fitness function to evaluate their performance in the solution domain.

Therefore, this paper documents the application of a genetic algorithm to determine an ideal set of parameters that accurately simulate the behaviour of the biogeochemical component (PISCES-v2) of an ocean model. The overall aim of this investigation is to illustrate that using computational intelligence techniques for parameter estimation in Earth system models is an effective approach, and to explore via a GA how this can be implemented. We also demonstrate how to embed a state-of-the-art ocean model as a prior step to the computation of the GA cost function by leveraging a workflow manager used to run climate





experiments operationally. Finally, we discuss how our approach can become the first step towards assimilating new kinds of observations into existing Earth system models.

## 2   Methodology

This section outlines the main methods used in this investigation. A test case of particulate organic carbon (POC) in the North Atlantic down to 1000 m is used. The observed data, explained in detail in 2.1, are obtained from autonomous ocean Argo floats. The model tested is the one-dimensional (depth) configuration of the ocean biogeochemical model PISCES-v2 (Aumont et al., 2015), a component of NEMO4 (Nucleus for European Modelling of the Ocean version 4), as outlined in 2.2.

The type of GA used is known as BRKGA (Goncalves and Resende, 2011). The outline of this method, including the crossover, is described in 2.3. We use the workflow manager Autosubmit (Manubens-Gil et al., 2016; Uruchi et al., 2021) to create a workflow that facilitates the various steps of the algorithm, as outlined in 2.4.

We run three test case experiments where the reference data are an output of a simulation with default parameters, another three where the reference data are observed data from three locations in the North Atlantic, and lastly a set of cross experiments.

Section 2.5 outlines the details of these experiments.

### 2.1   Biogeochemical Data

Our investigation focuses on the vertical profiles of in the North Atlantic subpolar gyre (Labrador Sea and Iceland Basin). The observed data were acquired by Argo floats deployed in the context of the international Argo program (Roemmich et al., 2019). Argo floats are autonomous drifting floats fitted with sensors that provide real time updates of ocean data. Over regular

intervals, each float rises from its drifting depth of 1000m to the surface, taking measurements in the process. When it reaches the surface, it transmits the measurements. Initially the Argo program focused on observing salinity and temperature but more recently has included biogeochemical measurements (Claustre et al., 2020). Our investigation focuses on the data of three floats deployed by the project remOcean (table 1). These floats took measurements every 1–3 days during times of high biological activity (i.e. phytoplankton blooms) and every 10 days for the rest of the year, with enhanced vertical resolution (Briggs et al.,

2020) compared to the regular Argo floats. The data were quality controlled (Organelli et al., 2017).

To successfully compare biogeochemical Argo data to model output, some challenges have to be overcome. First, biogeochemical Argo floats do not measure POC directly. Instead, POC is estimated from the measurement of optical backscattering by particles at 700 nm wavelength (bbp700). This measurement generally shows a linear relationship with POC, and appropriate conversion factors are available for several ocean regions including the subpolar North Atlantic. Here we used conversion

factors from Cetinić et al. (2012). Second, estimation of different POC size fractions can only be achieved when high vertical resolution measurements are available. Here we followed the method of Briggs et al. (2020). Briefly, this method separates the backscattering signal into baseline and spike signals which correspond, respectively, to small POC (sPOC) and big POC (bPOC), with a cutoff particle diameter of approximately 100 $\mu$m. Thus, sPOC corresponds to particles nominally smaller than 100 $\mu$m that are suspended or sink slowly, approximately at < 10 m d$^{-1}$ (Stemmann et al., 2004; Giering et al., 2016); big POC





(bPOC) corresponds to particles larger than 100 $\mu$m whose sinking rates are typically in the order of several tens or hundreds of

m d$^{-1}$. Finally, the trajectories of the drifting floats have to be matched to the model grid. Here, we treated the floats as "fixed",

with the location corresponding to the nearest grid point of NEMO that best represents the observed physical data from the

float. The reader is referred to the companion paper by Galí et al. (in prep.) for an extensive comparison between annual POC

time series from Argo floats and their PISCES 1D simulated counterparts, showing good agreement in the subpolar North

Atlantic.

| Location ID | Location | Longitude | Latitude | Year | Float WMO number |
|:---:|:---:|:---:|:---:|:---:|:---:|
| LAB1 | Labrador Sea | $-46.22$ | 57.20 | 2016 | 6901527 |
| LAB2 | Labrador Sea | $-54.85$ | 57.11 | 2014 | 6901527 |
| LAB3 | Labrador Sea | $-50.34$ | 56.31 | 2015 | 6901486 |

**Table 1.** Summary of the locations of the observations used to run the experiments. Each Argo float is identified by a World Meteorological
Organization (WMO) number

## 2.2   PISCES 1D and Parameters

PISCES-v2 (Aumont et al., 2015) is an OBGCM of intermediate complexity that represents the cycles of the main inorganic

nutrients (N, P, Si and Fe), carbonate chemistry, and organic matter compartments, including phytoplankton and zooplankton

organisms (two size classes each), dissolved organic matter, and particulate organic matter, making up 24 prognostic variables

or *tracers* in total. In PISCES-v2, detrital POC is divided into two size classes, with a nominal cutoff at 100 $\mu$m. Yet, it is

important to note that total POC as sampled in-situ is made up of detrital matter, phytoplankton and zooplankton. Therefore,

here we define sPOC as the sum of PISCES-simulated nanophytoplankton ("PHY"), microphytoplankton ("PHY2"), micro-

zooplankton ("ZOO") and small detrital POC ("POC"); bPOC is defined as the sum of PISCES-simulated mesozooplankton

("ZOO2") and big detrital POC ("GOC") (Gehlen et al., 2006). These idealized fractions show good correspondence with those

determined from BGC-Argo data (Galí et al., in prep.). Bacteria (heterotrophic prokaryotes) also contribute to the POC pool.

However, bacterial biomass is not a prognostic (interactive) tracer in PISCES-v2, and is implicitly included in the PISCES

tracer POC (Galí et al., in prep.). Therefore, although bacterial biomass is diagnosed in PISCES, it is not considered explicitly

in this study.

    Among the several tens of parameters that control biogeochemical process rates in PISCES-v2, here we focus on 9 param-

eters expected to strongly influence mesopelagic POC dynamics (table 2). These parameters control POC formation in the

surface productive layer through microphytoplankton mortality, gravitational POC fluxes, POC degradation rates, and inter-

ception of sinking POC by mesopelagic zooplankton. Preliminary tests also included the parameters *unass* and *unass2*, which

determine POC production from the unassimilated fraction of phytoplankton biomass ingested by zooplankton. However, they

were not included because these parameters have a strong impact on epipelagic ecosystem dynamics, which are beyond the

scope of our study. We refer the reader to the works of Aumont et al. (2015, 2017) for further information on the structure and

default parameter values of PISCESv2.



| Parameter | Definition | Default value | Range | Units |
|-----------|------------|---------------|-------|-------|
| *wchld* | microphytoplankton mortality | 0.01 | 0–0.05 | d$^{-1}$ |
| *wchldm* | microphytoplankton aggregation | 0.05 | 0–0.10 | d$^{-1}$ |
| *caco3r* | fraction of calcifying nanophytoplankton | 0.3 | 0–0.8 | unitless |
| *wsbio* | sPOC sinking speed | 2 | 0–10 | m d$^{-1}$ |
| *wsbio2* | minimum bPOC sinking speed | 50 | 0–250 | m d$^{-1}$ |
| *wsbio2max* | maximum bPOC sinking speed | 50 | 0–1000 | m d$^{-1}$ |
| *xremip* | specific fresh POC remineralization (0 degrees) | 0.035 | 0–0.10 | d$^{-1}$ |
| *grazflux* | flux-feeding cross-section | 3000 | 500–10000 | m$^2$ m$^{-3}$ (mol zoo. C)$^{-1}$ |
| *solgoc* | bacterial bPOC-to-sPOC solubilization | 0.11 | 0–0.5 | unitless |

**Table 2.** Definitions of the PISCES parameters included in the optimisation experiments, along with their default values, optimisation ranges, and units.

This investigation uses PISCES-v2 coupled with NEMO4 and configured for one spatial dimension (1D) and to run offline (Galí et al., in prep.; see also Llort et al. (2015)). The 1D configuration has the same vertical levels as the 3D configuration (in our setup, 75 levels of gradually increasing thickness —L75 vertical grid) but the horizontal grid is reduced to an idealized

domain of 3x3 cells. In this configuration, tracer concentrations change over the temporal and vertical dimensions as a result of local sources and sinks, vertical diffusion, particle sinking through the water column, and fluxes at the ocean-atmosphere boundary. PISCES computes the biogeochemical sources and sinks and the sinking of detrital particles at each "biological" time step (here set to 45 min, one-fourth of the NEMO time step). Then, the NEMO component TOP (Tracers in the Ocean Paradigm) calculates vertical diffusion using dynamical fields, which are precalculated in a previous NEMO run, with a time

step of 3 h. The 1D configuration does not allow for the advection of biogeochemical tracers, which would otherwise be computed by TOP in 3D runs. Simulations are spun up by repeating the same annual forcing over 4 years, and simulation year 5 is used for the comparison against observations. Annual cycles of pelagic biomass and detritus become identical from year 3 onward thanks, in part, to the nudging of nutrient concentrations towards the annual climatology below 300 m (Galí et al., in prep.).

Being one dimensional, the model only requires one computational core and runs at a speed of roughly one simulation year per minute on a supercomputer. This allows for multiple simulations to be run in parallel, hence making the 1D configuration an ideal test case for our investigation (Llort et al., 2015). The numerical parameters that will be constrained are stored in text files called namelists, and can be easily modified prior to each simulation without requiring recompilation. In the experiments (section 2.5), parameters were allowed to vary between 0 (as negative values are physically meaningless) and an upper bound

that exceeded the default values by between three- and 20-fold. The upper bounds, which are necessarily arbitrary, were chosen based on what we considered physically or biologically reasonable according to the experimental and modelling literature, whose review is beyond the scope of this paper.





## 2.3 Genetic Algorithm (GA)

A GA is a type of evolutionary algorithm used for optimisation that, in general, is analogous to natural selection in the sense that
a population of $p$ individuals are tested for their "strength" using a cost function, and weaker individuals get eliminated while
stronger individuals pass on their characteristics by pairing with other individuals to produce $\lambda$ offspring. In most applications
including this one, $p = \lambda$. A GA is considered a stochastic optimisation method, in the sense that there exists a balance between
elitist and exploratory behaviours, i.e., a more elitist algorithm will be less exploratory and vice versa. Being elitist in this sense
is the property of reaching an optimal solution with efficiency, and being exploratory refers to increasing the range of possible
solutions. Being exploratory is particularly important to ensure that the algorithm doesn't reach a local minimum of the cost
function by leaving some regions of the search space unexplored. The usual method of recombination in the GA is the crossover,
which is the action of two individuals from a generation producing offspring for the next. This is the primary discovery force
of the GA. In our case, an individual is a vector of floating point numbers that represent the values of the parameters. A
crossover occurs when two individuals are selected, and a new individual vector is created by taking a random combination
of components to the two individuals. In general, crossovers are intended to be elitist by ensuring that individuals with higher
strength are more likely to be chosen. This process is known as selective pressure. Another feature inspired by genetics is the
concept of mutations. The purpose of mutations is to make the algorithm more exploratory by randomly changing or perturbing
parts in individual members or adding randomly generated individuals to the population. This is usually done with a very small
probability, emulating transcription errors that occur within natural gene-passing. Once the crossovers are completed and the
new generation is made, their 'strength' is again measured and the process is repeated. This continues until a certain condition
is met. This can be once the value of the cost function of the strongest member reaches a certain value, or if no change is noted
after a certain number of generations, or simply after a predetermined number of generations.

### 2.3.1 Biased Random Key Genetic Algorithm (BRKGA)

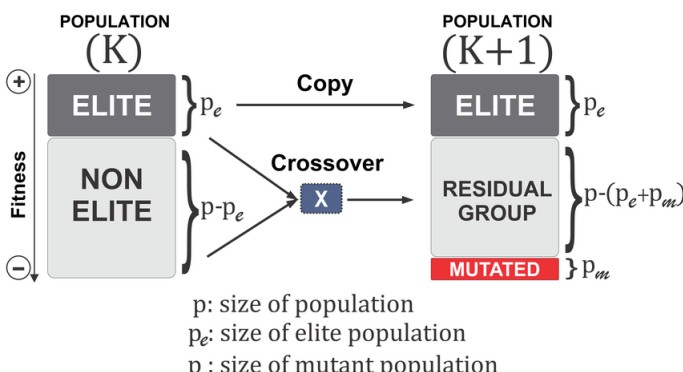

**Figure 1.** A visualisation of the BRKGA's process from one generation to another. (Júnior et al., 2020)





A BRKGA is a particular type of GA where each gene is a vector of floats rather than a bitstring which is typical of traditional GAs (De Jong et al., 1993). This is useful to address the issue of uneven distance between solutions, inherent to bitstrings, and appropriate for this problem because the set of parameters to be optimised can be treated as vectors. The behaviour of the BRKGA can be adjusted by changing the so-called metaparameters (figure 1) that are described below. Initially, $p$ sets of parameters are generated from random using a uniform distribution with appropriate bounds (section 2.2). At each generation, the vectors are partitioned into a set of size $p_e$, where $p_e < p/2$, which contains the elite population, that is, the individuals with the best score. These are passed directly to the next generation. The remainder of the vectors are partitioned into the non-elite set. Next, $p_m$ sets of randomly generated sets of vectors are introduced into the population as 'mutants', and passed directly onto the next generation in order to make the algorithm more exploratory, performing the same role as mutations in standard GAs. The set of vectors of the next generation is completed by generating $p - (p_e + p_m)$ vectors by crossover. A crossover in this case is a method used to generate a new vector set by selecting two parents at random and then each element of the new vector is randomly picked from one of the two parents. In a normal Random Key GA (RKGA) the parents are selected completely from random from the whole of the previous set of parameters, with a 0.5 probability of an element coming from either parent. However in a BRKGA one 'parent' vector comes from the elite set and the other from the non-elite set. In addition, the probability of an element coming from an elite parent is determined by $\rho$, where $\rho > 0.5$. This has shown in previous investigations to cause faster convergence to an optimal solution (Goncalves and Resende, 2011). Finally to make the algorithm more exploratory, after the crossover is completed all of the values are slightly perturbed to allow the exploration of values close to those of the elite vectors. It is worth noting that this slight perturbation may allow the parameters to evolve beyond their initial range - given that the parameter ranges are also not well constrained, this allows the algorithm to explore the possibility of finding optimal values outside the given range, however, the feasibility of the values is at the discretion of the user.

### 2.3.2 Cost Function

Deciding on an ideal cost function to measure the misfit between the results of each simulation and the observed data requires a number of considerations. In this case, the limitations of the model itself and the particular properties of the data need to be taken into account. An important model limitation is that there exists inherent physical biases, and in some cases, uncertainties in the conversion factor between the model variable and its observed counterpart. In addition, we wish to compare trends, in particular the seasonality of the data. For this, simply calculating the difference between observed data and simulated outputs, or bias, is not sufficient.

To ensure sensible fitting, in addition to bias, the correlation and the normalised standard deviation need to be considered. The Root Mean Square Error, RMSE, is a widely used parameter in this type of investigation, however in certain cases it has been found to reward reductions in model variability, for example over the seasonal cycle (Jolliff et al., 2009). An alternative metric known as the ST score is used. This is defined as:

$$ST = \sqrt{Bias_m{}^2 + S_3{}^2} \tag{1}$$





$Bias_m$ of an individual simulation is defined as its mean bias (over all data points) divided by the mean bias of the individual with the highest bias in the particular generation, that is

$$Bias_m = \frac{Bias_i}{Bias_{max}} \tag{2}$$

230 $S_T$ is a function of normalised standard deviation, $\sigma$, and correlation, $R$. Jolliff et al. 2009, (Jolliff et al., 2009) tests this particular cost function using bio-optical data, generally characterised by log-normal or similar right-skewed distributions that reflect the exponential growth and decay of plankton organisms. For this reason a normal logarithmic scale is used, a choice that is supported by preliminary experiments where the GA performance with linear- vs. log-space statistics was evaluated. Jolliff et al. (2009) state various possible formulae. Since it is of high importance to correctly determine seasonality in this 235 investigation and in this field in general, it is most sensible to choose a cost function that prevents situations where normalised standard deviation and bias are rewarded at the expense of correlation. Considering the three described options, $S_3$ served this purpose most appropriately:

$$S_3 = 1.0 - \left( e^{-\frac{(\sigma - 1.0)^2}{0.18}} \right) \left( \frac{(1+R)}{2} \right) \tag{3}$$

The philosophy behind ST is close to that of Taylor diagrams (Taylor, 2001), with the advantage of being a single value.

240 ## 2.4 Workflow

Running a GA requires performing a number of iterations until a termination condition is achieved. This does not represent a technical challenge if the fitness function can be calculated directly from the generation members. However, in some cases such as the one presented in this work, an external model is responsible for calculating the result that will be the input to the cost function. As a consequence, the need for parallel execution and management of many different jobs requires using tools 245 called workflow managers or meta-schedulers, which are commonly used to run ensemble experiments with climate models. Here we use a state-of-the-art workflow manager called Autosubmit (Manubens-Gil et al., 2016). Autosubmit is developed with models for Earth Sciences in mind, and is typically used to run complex simulations composed of multiple different jobs executed in one or multiple clusters via SSH connection. Autosubmit can handle the task submissions fulfilling dependencies between individual jobs and managing failures with minimal user intervention, providing tools to monitor (Uruchi et al., 250 2021) the experiment execution. In addition, it allows multiple jobs to run simultaneously in parallel or packed in macro-jobs ("wrappers") by automatically allocating the required computing resources.

Autosubmit experiments are hierarchically composed of start dates, members and chunks. A single experiment can run different start dates, that can be divided into members, in which each member contains an individual simulation. This feature was added to facilitate ensemble forecasts. In addition, each member is usually divided into different sequential chunks in 255 order to save checkpoints of the model state in regular intervals. With these features, Autosubmit has the ability to run multiple





members in parallel and therefore is suitable to run a GA in which there are different individuals in the same generation. This allows the size of the experiment to be adjusted easily and many different quantities of population and generations to be tested with ease. This particular use of Autosubmit to facilitate multiple instances of a computational model in a GA is a novel one. One shortcoming of this, however, is that the workflow size is strictly static and there does not exist a feature in which the workflow stops after certain conditions are met. This means the only viable stopping condition of the GA is after a predetermined number of generations, otherwise the stopping condition would have been if no evolution is observed after a certain number of generations.

Our particular workflow consists of three different types of job. The first is the initialisation of the experiment and is only run once at the very beginning of the experiment. The second is the simulation, ran once per individual in parallel in each generation. Finally the post processing, which includes the crossover, is ran once per generation.

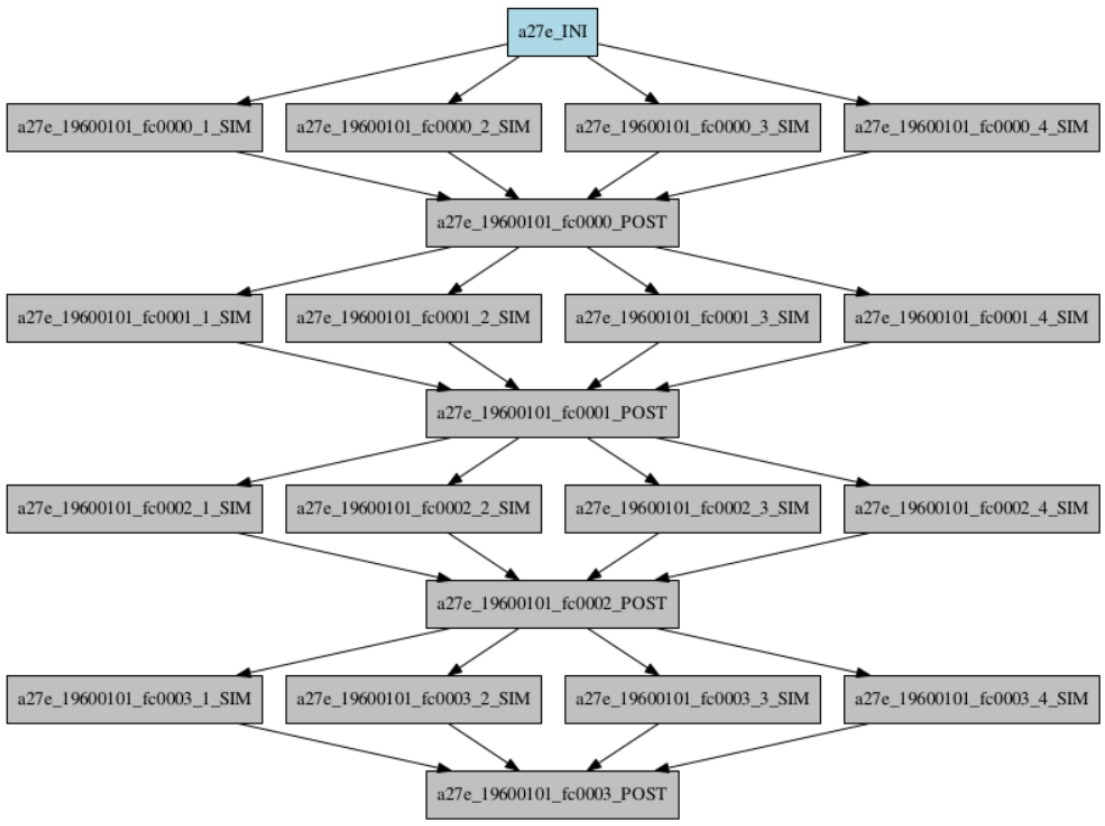

**Figure 2.** A toy example Autosubmit workflow with 4 population and 4 generations.

### 2.4.1 Initialisation

The initialisation script starts by setting up the directory in which the simulations are run by copying the executable of the model and the necessary input files into it. The domain files, which are by default global are then cropped into the appropriate





3x3 domain, depending on the location. Next the simulation is ran with default parameters, and certain statistical measurements

between its output and the observed data are taken that are necessary for post processing and calculation of the cost function. Finally, this script also generates the initial set of vectors from random to be tested.

### 2.4.2 Simulation

The second script, which runs $p$ times at each generation in parallel, starts by setting up the environment for each simulation. Then reads its corresponding vector from the generated set, edits the namelists to contain the updated parameters. Afterwards

the simulation is run. Then the statistics used to compute the cost function 2.4.1 are calculated.

### 2.4.3 Crossover

The final script runs once per generation after all simulations of the respective generation are completed. Firstly, it reads the statistics calculated after each simulation and uses them to calculate the ST score of sPOC and bPOC. It then ranks each of the simulations according to the sum of the two ST scores. Then it performs the crossover as described in 2.3.1 to produce a new

set of parameters in the same format so that it can be read by the following generation's simulation scripts.

### 2.5 Experiments

To investigate the potential of the GA, different sets of experiments are run. Each set contains 5 experiments (to test consistency and robustness) with distinct and randomly generated initial populations, with 100 individual simulations over 100 iterations. Their details are summarised in table 3.

| Experiment Set | Ref. Data | No. Parameters | Algorithm | Location |
|:---:|:---:|:---:|:---:|:---:|
| **D9** | Default Sim | 9 | GA | LAB1 |
| **D5** | Default Sim | 5 | GA | LAB1 |
| **D5_rand** | Default Sim | 5 | RS | LAB1 |
| **O5_LAB1** | Observed | 5 | GA | LAB1 |
| **O5_LAB2** | Observed | 5 | GA | LAB2 |
| **O5_LAB3** | Observed | 5 | GA | LAB3 |

**Table 3.** A summary of the experiments run using the workflow.

Initially, we determine the capabilities of the GA by testing how well it can find a known set of parameters. To do this, experiment sets **D9**, **D5** and **D5_rand** are run using the output of a simulation with default parameters as the reference data at location LAB1. In set **D9**, 9 parameters are tested to check which ones can be constrained from sPOC and bPOC data. This leads us to select 5 parameters, which are tested in set **D5**, additionally giving us an indication of how the method behaves when different sizes of parameter sets are used. Alongside this, set **D5_rand** is a random search (RS) algorithm — experiments

that use a different optimisation algorithm similar to the BRKGA except that only the best parameter set is passed to the next





generation and every other set is randomly generated. This is to determine if our method based on BRKGA can find an optimal parameter set quicker and more consistently than a trial-and-error approach with random parameter sets.

Experiment set **O5_LAB1** uses the GA as intended, where the reference data are observed data from LAB1 and outputs are analysed. This is further compared with experiment sets **O5_LAB2** and **O5_LAB3**, which are run in LAB2 and LAB3

respectively. This is to investigate how the results obtained reflect the wider region.

Finally, cross simulations are run, whereby a representative parameter set from each experiment sets **O5_LAB1**, **O5_LAB2** and **O5_LAB3** is selected to run a single simulation in the other two locations. This is to further check how robust the GA is and if the parameter sets produced are representative of the region. In fact, a certain homogeneity is expected across the three locations because of their similar physical and biogeochemical properties. The GA not capturing this homogeneity would

suggest the tool is compensating for other errors in the attempt to minimize the cost function resulting in an overfitting of the optimal parameter set.

## 3 Results

### 3.1 Default Data

#### 3.1.1 Nine parameters

The evolution of the optimal sets of parameters in experiment set **D9** is presented in figure 3. Table 4 shows the optimal parameter set in the final generation of each experiment. Figures 4 and 5 present the cost function of each optimal set per iteration and their corresponding statistics of the sPOC and bPOC, and table 5 shows the statistics of the optimal parameter set at the end of each experiment. In all cases, most of the evolution occurs within the first 10 to 20 generations. This is evident from all figures, as the cost function decreases rapidly towards zero and the optimal sets of parameters in all experiments

fluctuate greatly initially before remaining at similar values for the remainder of the experiment. The spread of the values to which the parameters tend to converge varies strongly from one parameter to another. When a parameter evolves towards its corresponding default value in a consistent manner across the 5 replicate experiments, this suggests it can be constrained from POC variables with greater confidence, and should be considered when trying to optimise the model against observed data. This is evident with *wsbio* and *xremip*, which return immediately to the default value in every experiment. Other parameters,

like *wchld*, *wchldm*, *grazflux*, *solgoc* and *wsbio2*, show larger optimisation uncertainty but converge to within ±50% of the default value in most cases. On the other hand, if the optimal values for a parameter in each of the 5 experiments differ from each other and the default, this suggests that they cannot be constrained from POC variables in the 0–1000 m domain. This can be seen with *wsbiomax* and *caco3r*. Differences between experiments are also indicative of tradeoffs between parameter changes and their impact on the cost function. For example, in experiment a274, its distinctly lower *xremip* value, along with

its lower correlation and higher RMSE of bPOC (figure 5), suggest that it had optimised sPOC quickly at the expense of bPOC, causing the GA to get trapped in a local minimum as indicated by the higher overall cost.

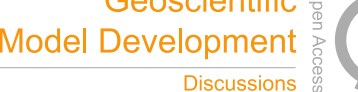

The results of experiment **D9**, plus additional analyses that we report in the Supplemental Information (SI), provided the criteria to select the 5 parameters that were used in subsequent PO experiments. Quite obviously, the sPOC sinking speed, *wsbio*, and the fresh POC specific remineralization rate, *xremip*, were selected owing to their rapid and robust convergence to
the expected values. In addition, *wchld*, *wchldm* and *grazflux*, which showed vacillating convergence behaviour, were selected owing to their important role in POC budgets in the study area. In particular, the microphytoplankton (diatoms) natural mortality *wchld* and aggregation *wchldm* control detrital POC formation as phytoplankton blooms collapse. The zooplankton flux-feeding cross section, *grazflux* controls the vertical attenuation of the gravitational bPOC flux in the upper mesopelagic, transferring a fraction of it to sPOC. Our POC budget calculations (figures S1 and S2) and sensitivity analyses (figures S3-S5) indicate
that *wsbio*, *xremip*, and *grazflux* exerted the strongest control on mesopelagic sPOC and bPOC concentrations in our study area. Additional experiments (not shown) were run with *solgoc* and other parameters, supporting the choice of the previous 5 parameters.

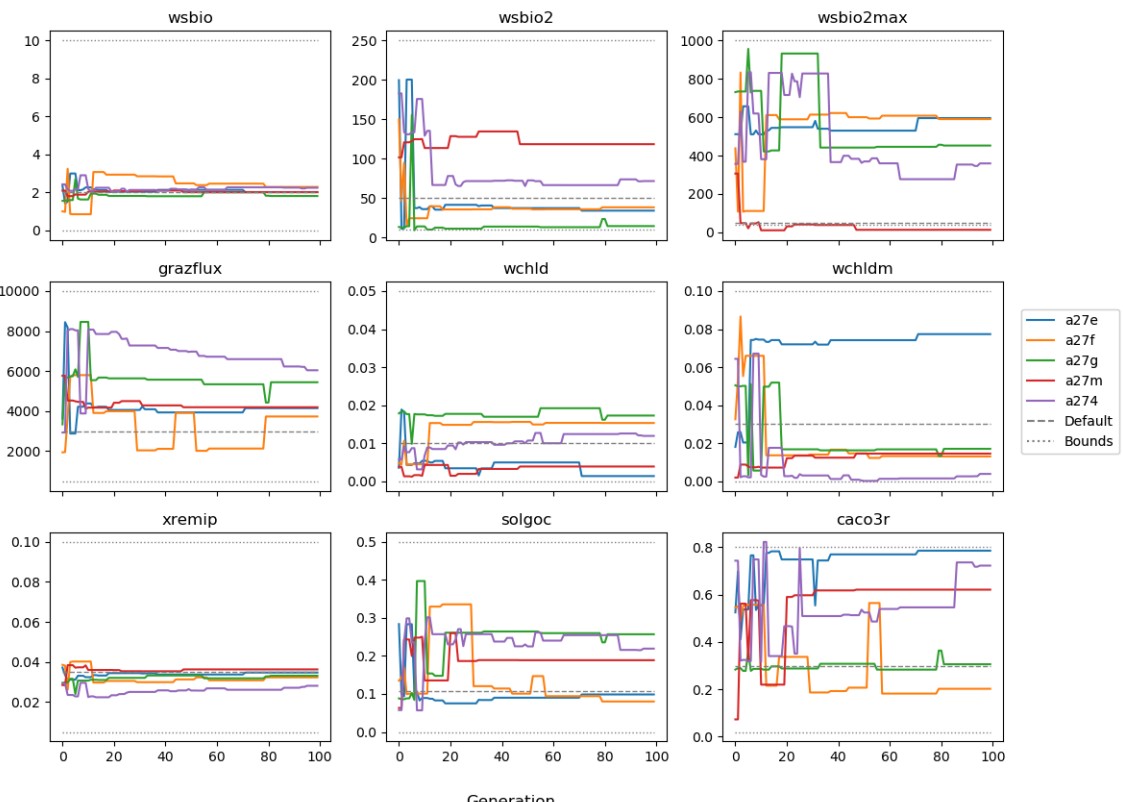

**Figure 3.** Evolution of each generation's optimal set of parameters in experiment set **D9**

| Parameter | default | a27e | a27f | a27g | a27m | a274 |
|-----------|---------|------|------|------|------|------|
| *wchld* | 0.01 | 0.001 | 0.015 | 0.017 | 0.004 | 0.012 |
| *wchldm* | 0.03 | 0.078 | 0.013 | 0.017 | 0.015 | 0.004 |
| *caco3r* | 0.3 | 0.79 | 0.20 | 0.31 | 0.62 | 0.72 |
| *wsbio* | 2 | 2.01 | 2.28 | 1.80 | 2.00 | 2.24 |
| *wsbio2* | 50 | 34 | 38 | 15 | 118 | 72 |
| *wsbio2max* | 50 | 595 | 590 | 452 | 13.6 | 359 |
| *xremip* | 0.035 | 0.035 | 0.053 | 0.033 | 0.036 | 0.028 |
| *grazflux* | 3000 | 4140 | 3730 | 5440 | 4190 | 6040 |
| *solgoc* | 0.11 | 0.10 | 0.08 | 0.26 | 0.19 | 0.22 |

**Table 4.** The parameter sets of the final generations with the lowest ST score of experiment set **D9**, and the default parameter set.

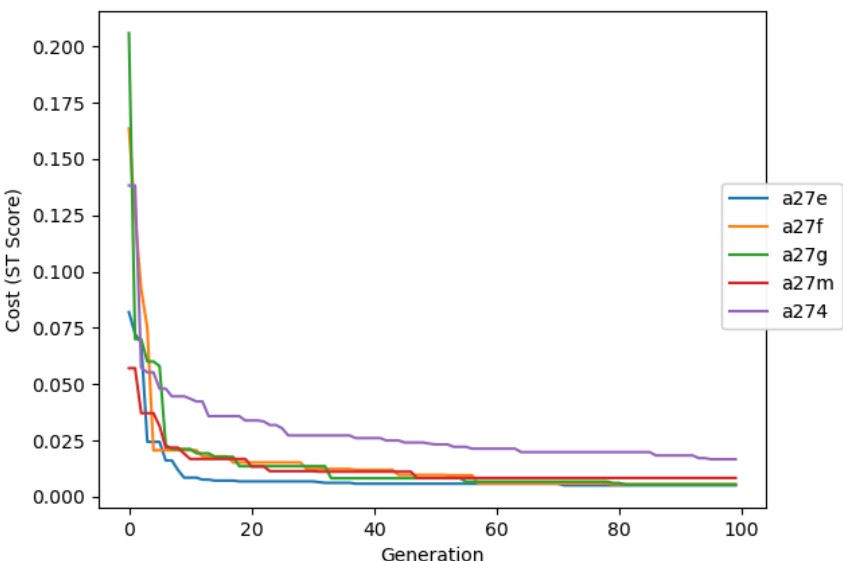

**Figure 4.** Evolution of each generation's lowest ST score for experiment set **D9**.




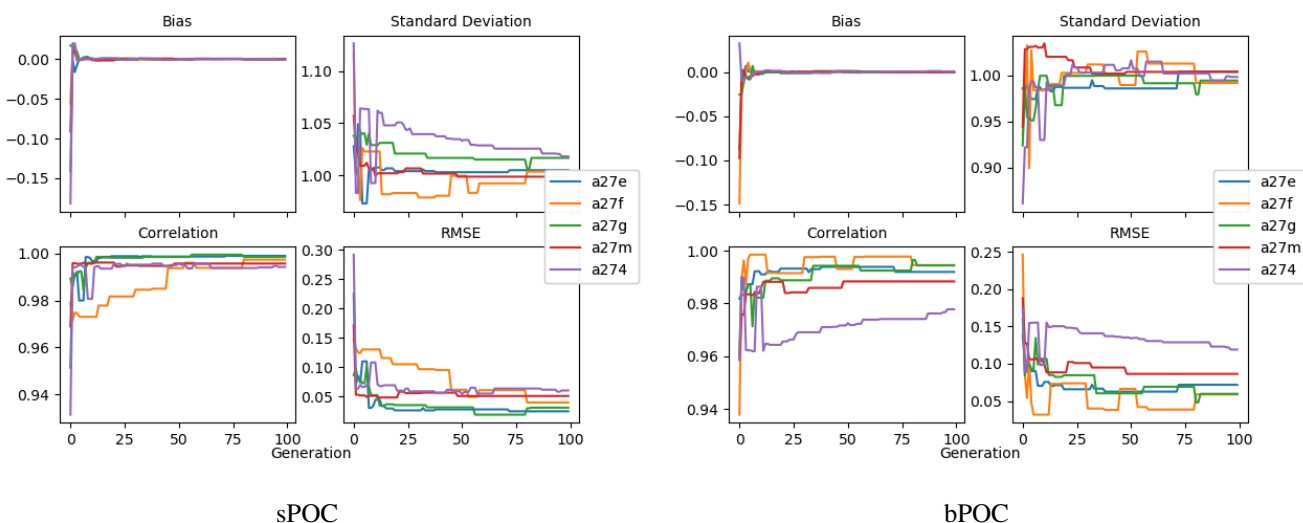

sPOC                                                    bPOC

**Figure 5.** Evolution of each optimal generation's bias, normalised standard deviation, correlation and RMSE of experiment set **D9**.

| | Stat | a27e | a27f | a27g | a27m | a274 |
|---|---|---|---|---|---|---|
| | Cost | 0.0051 | 0.0055 | 0.0055 | 0.0083 | 0.0166 |
| sPOC | Bias | 0.000188 | −0.000620 | −0.000165 | −0.000004 | −0.000691 |
| | StDev. | 1.0049 | 1.0035 | 1.0167 | 0.9987 | 1.0181 |
| | R | 0.9990 | 0.9975 | 0.9985 | 0.9957 | 0.9942 |
| | RMSE | 0.0253 | 0.0403 | 0.0315 | 0.0513 | 0.0608 |
| bPOC | Bias | 0.000013 | 0.000111 | −0.000167 | −0.000343 | 0.000245 |
| | StDev. | 1.0042 | 0.9918 | 0.9944 | 1.0039 | 0.9984 |
| | R | 0.9920 | 0.9945 | 0.9944 | 0.9884 | 0.9778 |
| | RMSE | 0.0718 | 0.0589 | 0.0596 | 0.0863 | 0.1189 |

**Table 5.** Statistical values for the sPOC and bPOC of the final parameter set of each experiment in set **D9**.

### 3.1.2 Five parameters

The following plots and tables analyse the results of experiment set **D5**. The evolution of the optimal parameter set of each
generation is presented in red in figure 6, and the final values of each parameter of each experiment is presented in table 6.
Figures 7 and 8 show the evolution of these experiments' statistics, and table 7 shows the statistics of the final generation.
When comparing these to those for experiment set **D9**, an all-round improvement is visible. In all cases, the parameters are
more consistent and are more likely to return to the default, and quicker. There is less indication of the experiments getting
stuck in a local minimum, i.e., producing a result that has the lowest cost function for a subset of the parameter space, while

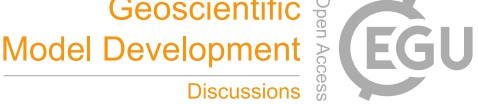

there exists a lower cost function elsewhere within the wider space. In all cases, the cost functions are lower and the rest of the statistics are also better. Preliminary experiments where only 3 parameters were optimised (not shown) yielded even faster and more robust convergence to the expected parameter values. These results suggest that with larger parameter sets, the GA requires a larger population and a larger number of generations to be effective. However, given the difference in the results of sets **D9** and **D5**, there is reason to believe that increasing the number of parameters in the GA does not increase the

dimensionality of the problem in the way that a brute-force approach would have. Finally, increasing the parameter set size increases the probability of the GA getting stuck in local minima while searching for the optimal set.

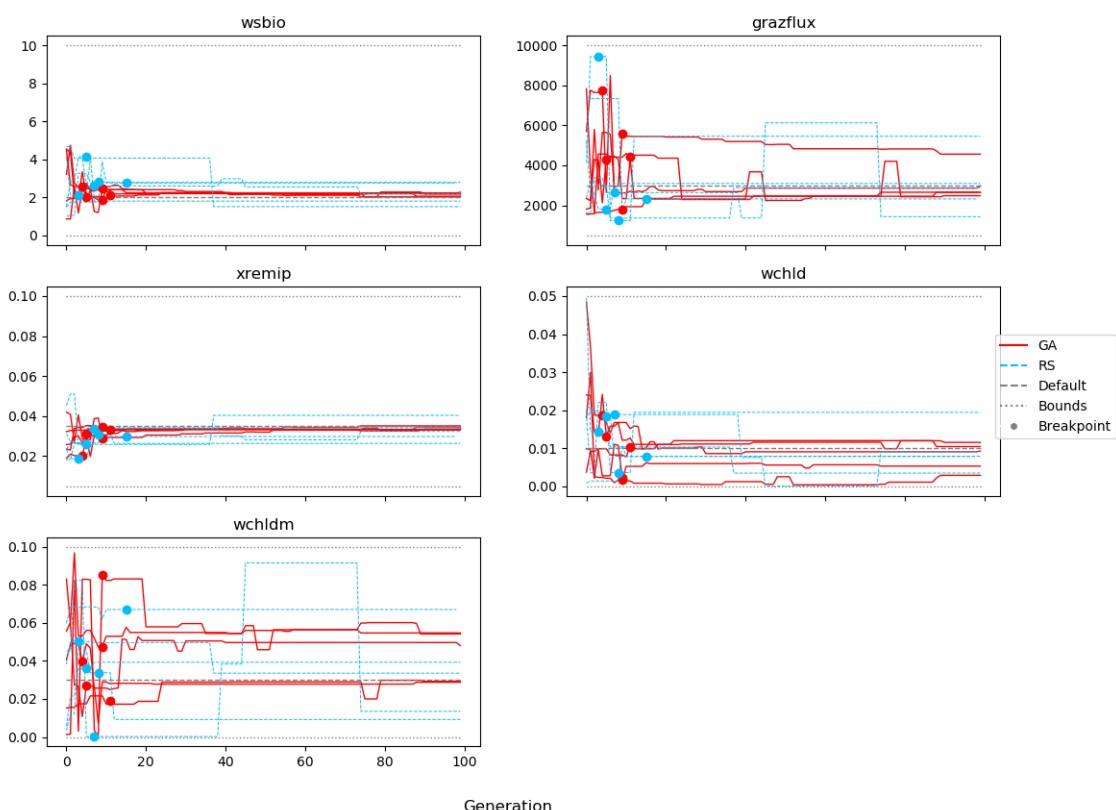

**Figure 6.** Evolution of each generation's optimal set of parameters for experiment set **D5** and set **D5_rand**



| Parameter | default | a27e | a27f | a27g | a27m | a274 |
|---|---|---|---|---|---|---|
| *wchld* | 0.010 | 0.0104 | 0.005 | 0.012 | 0.003 | 0.009 |
| *wchldm* | 0.030 | 0.029 | 0.055 | 0.029 | 0.054 | 0.048 |
| *wsbio* | 2 | 2.05 | 2.02 | 2.17 | 2.18 | 2.26 |
| *xremip* | 0.035 | 0.035 | 0.035 | 0.034 | 0.033 | 0.034 |
| *grazflux* | 3000 | 2651 | 2652 | 2478 | 4557 | 2969 |

**Table 6.** The parameter sets of the final generations with the lowest ST score of the experiments in set **D5**, and the default parameters.

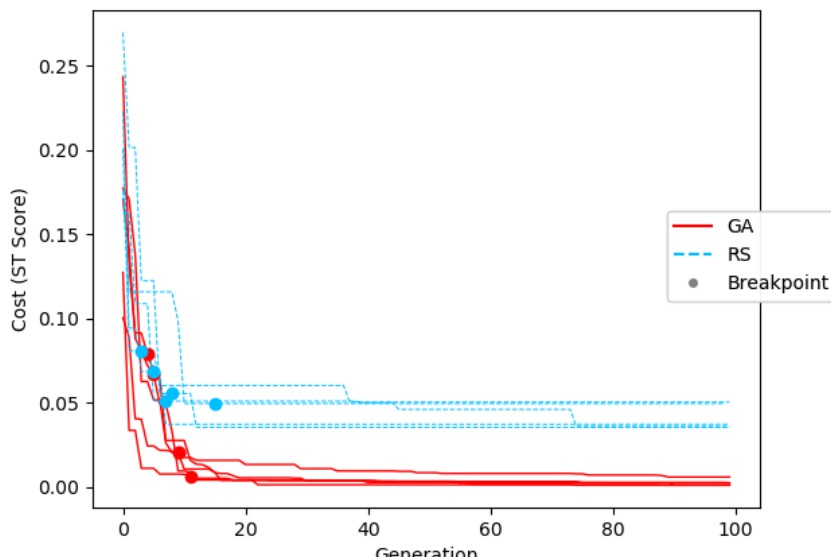

**Figure 7.** Evolution of each generation's lowest ST score in set **D5**.



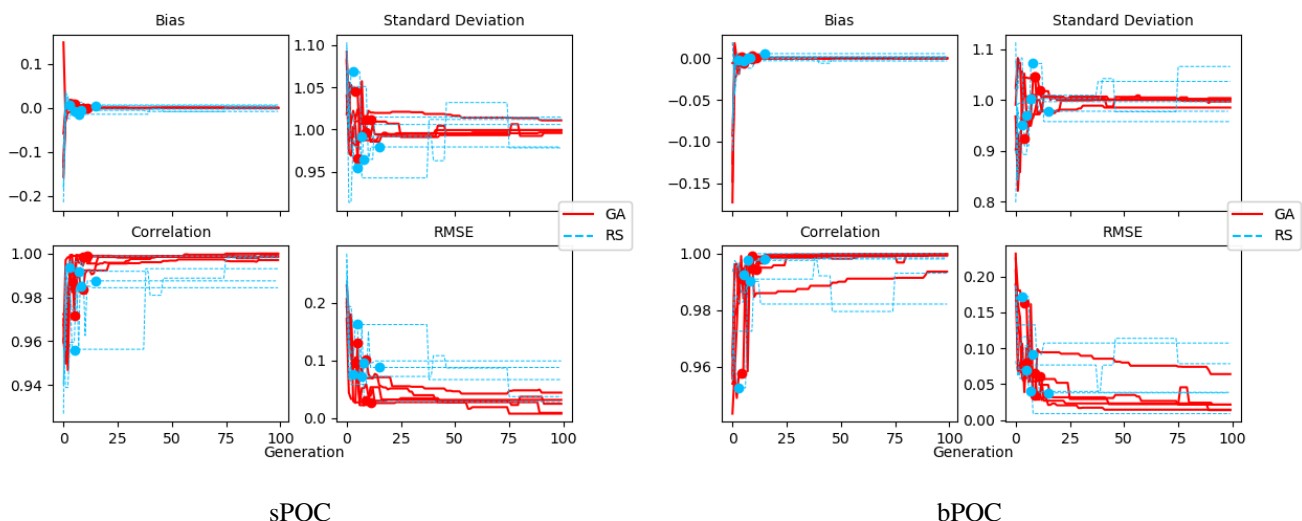

sPOC                                                bPOC

**Figure 8.** Evolution of each generation's optimal bias, normalised standard deviation, correlation and RMSE in experiment set **D5**.

| Stat | | a27e | a27f | a27g | a27m | a274 |
|------|------|------|------|------|------|------|
| Cost | | 0.001271 | 0.001244 | 0.001338 | 0.006088 | 0.002685 |
| sPOC | Bias | 0.000037 | −0.000207 | −0.000024 | −0.000422 | −0.000340 |
| | StDev. | 1.0997 | 0.9993 | 0.9960 | 1.0109 | 0.9994 |
| | R | 0.9999 | 0.9999 | 0.9990 | 0.9970 | 0.9984 |
| | RMSE | 0.0087 | 0.0073 | 0.0245 | 0.0438 | 0.0312 |
| bPOC | Bias | 0.000219 | −0.000015 | −0.000121 | 0.000094 | 0.000007 |
| | StDev. | 1.0035 | 1.0021 | 1.0002 | 0.9967 | 0.9849 |
| | R | 0.9997 | 0.9993 | 0.9993 | 0.9935 | 0.9998 |
| | RMSE | 0.0140 | 0.0216 | 0.0217 | 0.0641 | 0.0146 |

**Table 7.** Statistical values for the sPOC and bPOC of the final parameter set of experiment set **D5**.

### 3.1.3   Comparison with random search (RS)

In figures 6, 7 and 8, the evolution of the optimal set of parameters for each experiment in set **D5_rand** is plotted in blue. To compare the two sets of experiments (**D5** and **D5_rand**), the experiment with the median cost function is considered in

both cases, and the absolute difference between the final parameters and the default ones are calculated (table 8). The statistics of these two median experiments are presented in figure 9, alongside with the standard deviation of each statistic. The latter provides a metric for comparing the convergence robustness between (**D5** and **D5_rand**). Looking at both plots and the tables that compare sets **D5** and **D5_rand**, we can see that the GA outperforms the random search (RS) in almost every sense, with





few exceptions. The final parameter sets of the GA are more consistent than the RS, and all of the individual GA experiments
outperform the RS ones in the cost function and all of the statistics. The standard deviation of the statistics is higher for the
random search, providing further evidence that the convergence behaviour of the GA is more robust.

To further quantify the better efficiency of the BRKGA compared to the RS, we analyzed the evolution of each type of
experiment by means of a breakpoints analysis using the R package "segmented" (Muggeo, 2021). The analysis was performed
on the evolution of the cost function of the best parameter set of each experiment, as shown in figure 7. For each experiment,
we searched the generation number when the evolution of the cost function changed its decreasing trend and started to flatten.
This generation number, hereafter breakpoint, was found by splitting the evolution of the cost function into two segments,
such that they could be fitted with two optimal linear regressions that minimized the fit error. We then compared the values
of the cost function, statistics and model parameters at the breakpoints in each type of experiment. Both the **D5** and **D5_rand**
experiments reached the breakpoints with a similar speed, in 5–15 generations (average of 8). However, the mean and median
costs at the breakpoints were threefold lower in **D5**, corresponding to better statistics (figure 8) and parameter values closer
to their final optimised values after 100 generations (figure 6). Moreover, the **D5** experiments needed only 3-4 generations to
reach the cost of the **D5_rand** at the breakpoint. This analysis illustrates the greater efficiency of the BRKGA compared to the
RS. Breakpoint detection can be used to stop GA experiments in an adaptive manner, thus saving computation.

| Parameter | default | \|GA-default\| | \|RS-default\| |
|---|---|---|---|
| *wchld* | 0.01 | 0.0029 | 0.0043 |
| *wchldm* | 0.03 | 0.013 | 0.005 |
| *wsbio* | 2 | 0.14 | 0.48 |
| *xremip* | 0.035 | 0.004 | 0.018 |
| *grazflux* | 3000 | 558 | 984 |

**Table 8.** Absolute differences between the final parameter set of the median experiments of sets **D5** and **D5_rand** and the default parameter
set.





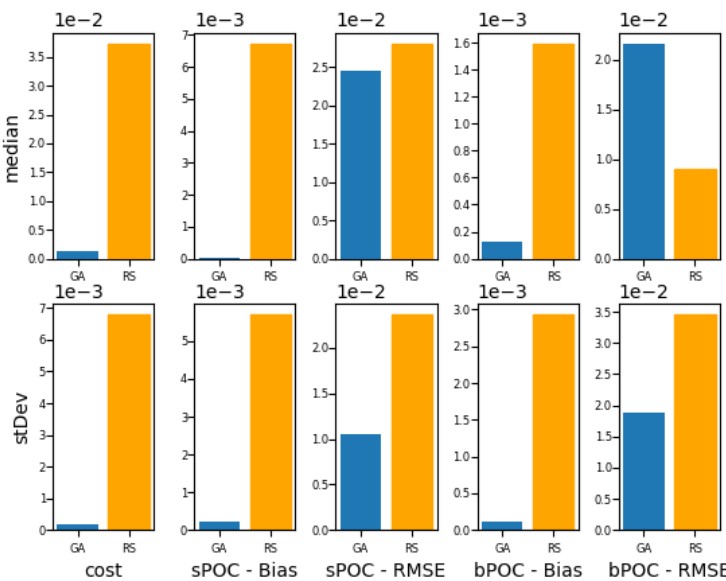

**Figure 9.** Comparison of the meta-analyses of experiment sets **D5** and **D5_rand**. The top row, 'median', compares the statistics of the experiments with the median cost function in each case. The bottom row, 'stDev', is the standard deviation of all experiments of each statistic in each case.

## 3.2 Observed Data

### 3.2.1 Labrador Sea

Figure 10 shows the evolution of the optimal set of parameters in each generation of experiment set **O5_LAB1**. Table 9 gives the final values of these parameters. We can see that in all 5 experiments, the parameters *wsbio*, *grazflux* and also *xremip* to a lesser degree, behave consistently, which is in line with how they behave in **D5**. Often the parameters move towards the extreme bounds of the initial range, and in some cases, beyond it. This is due to the slight perturbation of parameters after the crossover stage (section 2.3.1). The results also illustrate the interdependence between the parameters, such that a decrease in *wsbio* leads to an increase in certain others, a pattern that will be analyzed in the Discussion. The rapid evolution at the beginning is also evident in the large drop in the cost function that happens during the first 10 generations (figure 11), with breakpoints detected between generations 4 and 14. As expected, however, the cost function is overall higher than those of the experiments against the default outputs. From figure 12 we can see that most of the statistics improve very quickly at the start. Looking at table 10, it is noticeable that the statistics for the bPOC are generally worse than the sPOC, and hence make the larger contribution to the overall ST score. In figure 10 it can be seen that parameter values at the breakpoints are very close to, or indistinguishable from, their final values. Figures 13 and 14 are Hovmöller plots of the POC concentration profiles over the annual cycle for experiment **O5_LAB1** (observed, default model, and optimised model). Also included are





the deviations of the default and optimised outputs with respect to the observed data. With the sPOC, the improvement is
particularly noticeable in the elimination of the sPOC sinking plumes in the upper mesopelagic. Whereas mean biases are
generally reduced, patches with positive/negative biases remain at different times and depths after optimisation, which is also
reflected in the small improvements in correlation. It must be noted, however, that correlation coefficients were as high as 0.96
(0.85) for sPOC (bPOC) with the default parameters and thus difficult to improve. Further reduction in the bPOC misfit could
have been impeded by the noisier nature of the observed bPOC data.

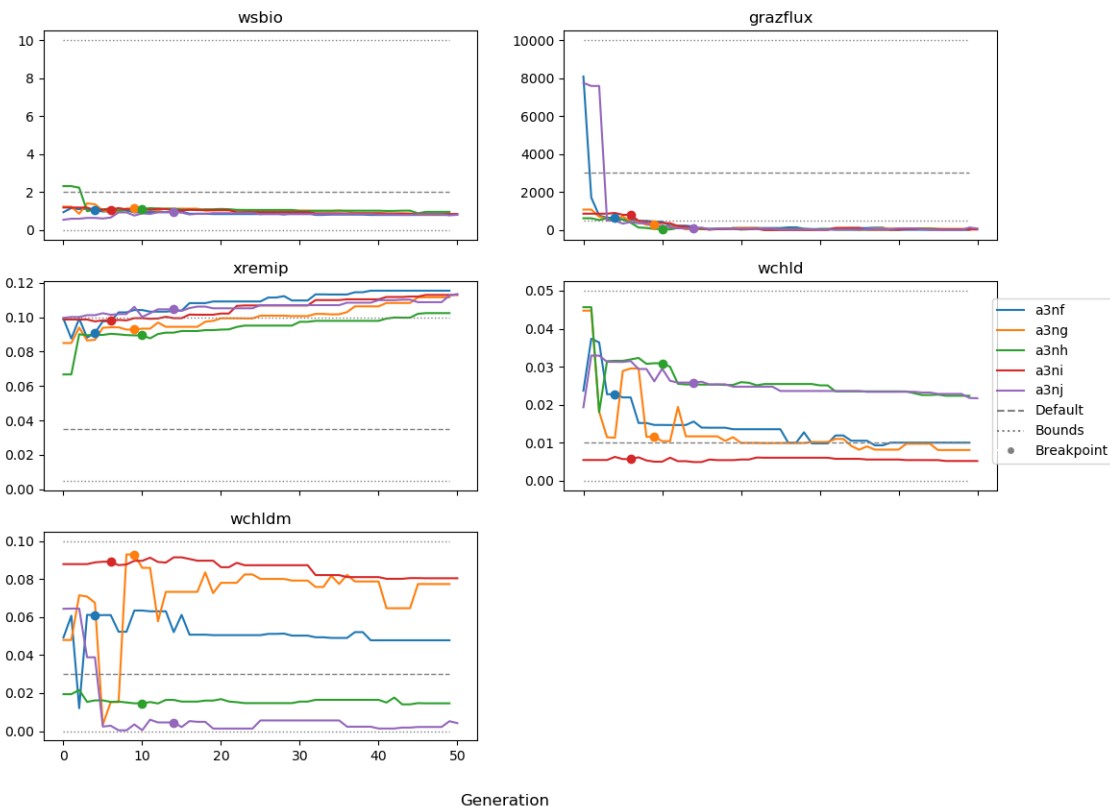

**Figure 10.** Evolution of each generation's optimal set of parameters for experiment set **O5_LAB1**.



| Parameter | default | a3nf | a3ng | a3nh | a3ni | a3nj |
|-----------|---------|------|------|------|------|------|
| *wchld*   | 0.010   | 0.010 | 0.008 | 0.022 | 0.005 | 0.022 |
| *wchldm*  | 0.030   | 0.047 | 0.077 | 0.002 | 0.081 | 0.004 |
| *wsbio*   | 2       | 0.79  | 0.81  | 0.95  | 0.84  | 0.80  |
| *xremip*  | 0.035   | 0.116 | 0.112 | 0.103 | 0.113 | 0.114 |
| *grazflux*| 3000    | 10.3  | 60.9  | 26.2  | 27.   | 77.3  |

**Table 9.** Evolution of each generation's optimal bias, normalised standard deviation, correlation and RMSE of experiment set **O5_LAB1**.

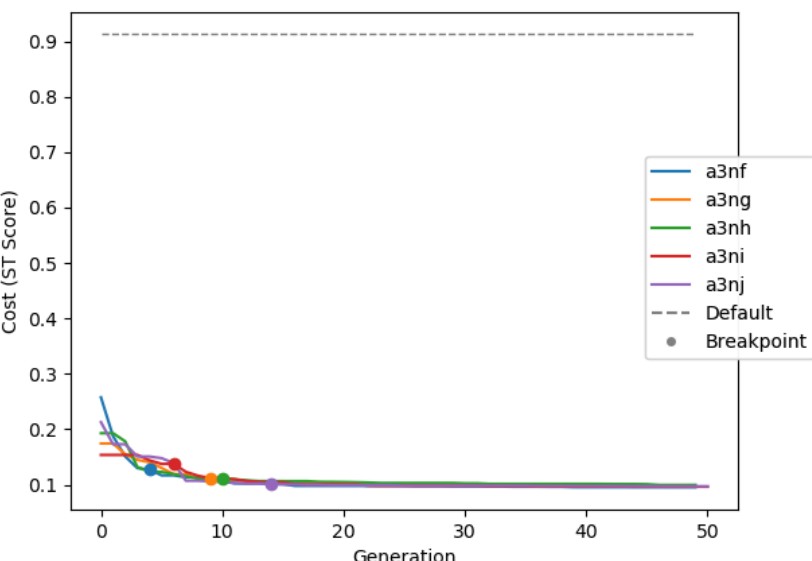

**Figure 11.** Evolution of each generation's lowest ST score of experiment set **O5_LAB1**. *Default* is the cost function of the default simulation.



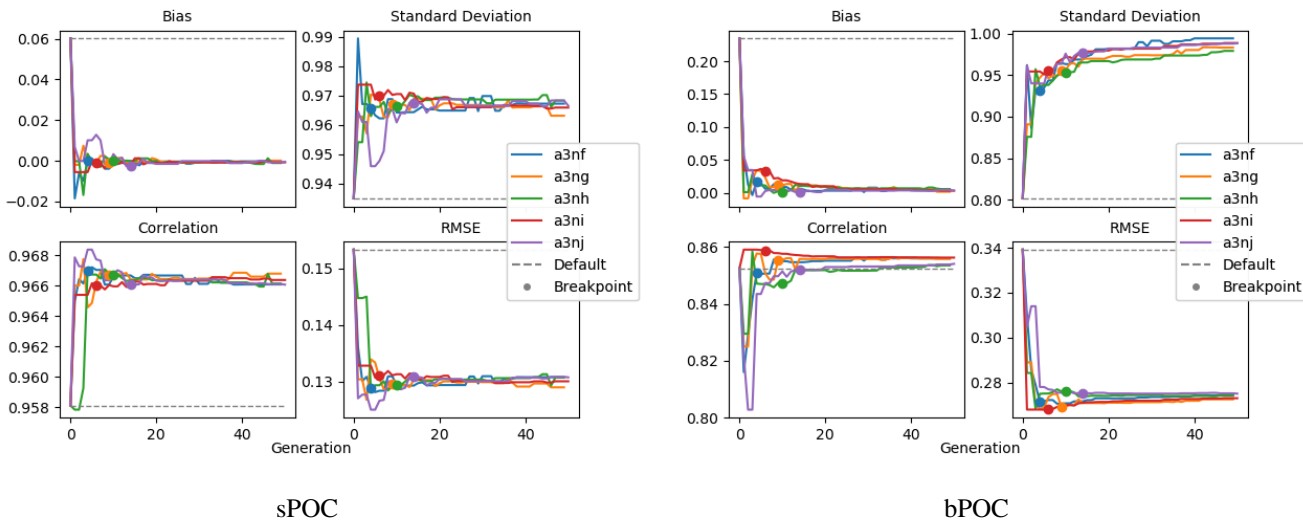

sPOC                                                         bPOC

**Figure 12.** Evolution of each generation's optimal bias, normalised standard deviation, correlation and RMSE of experiment set **O5_LAB1**.



| Stat | | a3nf | a3ng | a3nh | a3ni | a3nj |
|---|---|---|---|---|---|---|
| Cost | | 0.0957 | 0.0977 | 0.0997 | 0.0965 | 0.0976 |
| sPOC | Bias | −0.000246 | 0.000104 | −0.001048 | −0.000754 | −0.000450 |
| | StDev. | 0.967 | 0.963 | 0.968 | 0.966 | 0.967 |
| | R | 0.966 | 0.967 | 0.966 | 0.966 | 0.966 |
| | RMSE | 0.131 | 0.129 | 0.131 | 0.130 | 0.131 |
| bPOC | Bias | 0.00394 | 0.00193 | 0.00591 | 0.00348 | 0.00341 |
| | StDev. | 0.994 | 0.983 | 0.979 | 0.989 | 0.988 |
| | R | 0.856 | 0.856 | 0.854 | 0.856 | 0.854 |
| | RMSE | 0.274 | 0.273 | 0.274 | 0.273 | 0.275 |

**Table 10.** Statistical values for the sPOC and bPOC of the final parameter set of each experiment in set **O5_LAB1**.

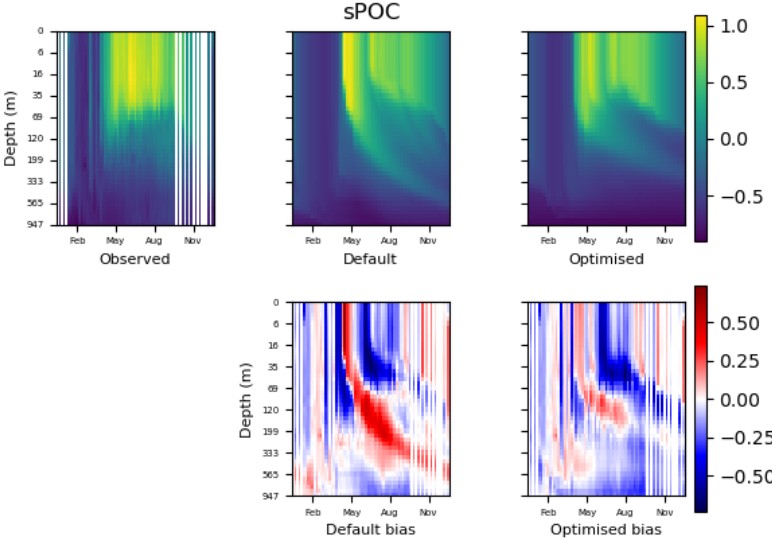

**Figure 13.** Top: data plots of sPOC in log scale for (L-R): observed data, the default parameter set's model output, and the optimised parameter set's model output. Bottom: the biases between the model outputs and observed data for the default parameter set (L) and optimised parameter set (R)



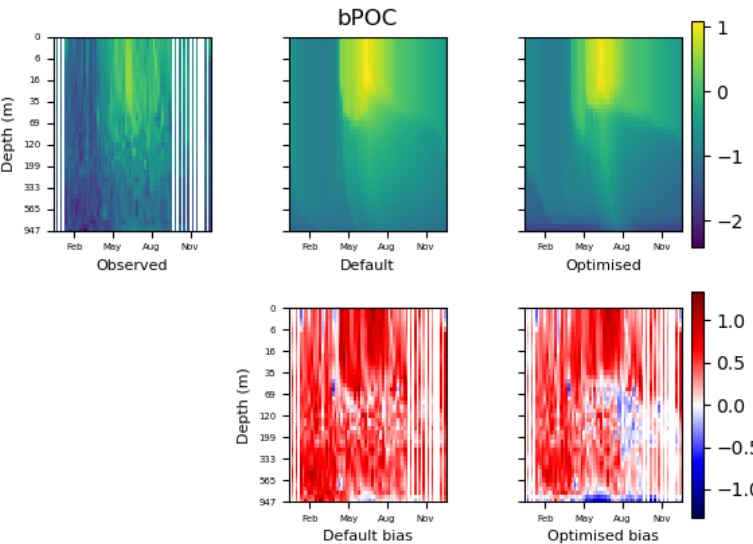

**Figure 14.** Top: data plots of bPOC in log scale for (L-R): observed data, the default parameter set's model output, and the optimised parameter set's model output. Bottom: the biases between the model outputs and observed data for the default parameter set (L) and optimised parameter set (R)

### 3.2.2 Experiments in other locations and cross testing


The experiments producing the median cost function for each set **O5_LAB1**, **O5_LAB2** and **O5_LAB3** are presented in table 11. We can see that the results are fairly consistent with each other, albeit some minor differences (for example wsbio in **O5_LAB1** and grazflux in **O5_LAB2**), indicating that the genetic algorithm behaves consistently from a regional perspective. This consistency is further confirmed when cross simulations are performed on the results. These cross simulations are per-

formed by using the parameter set produced for one location to run single simulations at the other two locations. The bias and correlation of sPOC and bPOC between the outputs of these simulations and the respective observed data are calculated. These statistics, along with the bias and correlation of the simulations with default parameters, are presented in tables 12 (sPOC) and 13 (bPOC). In all cases, when a parameter set obtained from one location is applied to another, the outputs show reasonable consistency. For bPOC, all cross tests show a substantial improvement in bias with respect to the default outputs and very little

improvement - if any - with correlation, which is consistent with the outputs from the original location. There are indications of consistency with sPOC, with nearly all showing an improvement with correlation, but it is less clear. This could be due to the default outputs' biases already being very low and their correlation being very high.



| Parameter | default | Labrador 1 | Labrador 2 | Labrador 3 |
|-----------|---------|------------|------------|------------|
| *wchld* | 0.010 | 0.0217 | 0.0392 | 0.0203 |
| *wchldm* | 0.030 | 0.0042 | 0.0358 | 0.0757 |
| *wsbio* | 2 | 0.795 | 0.179 | 0.008 |
| *xremip* | 0.035 | 0.114 | 0.094 | 0.078 |
| *grazflux* | 3000 | 77.3 | 9.8 | 72.2 |

**Table 11.** The final parameter sets of three genetic algorithm experiments ran in three locations, along with the default.

| Location | Parameter Set | | | |
|----------|---------|------------|------------|------------|
| | default | Labrador 1 | Labrador 2 | Labrador 3 |
| Labrador 1 | 0.0602, 0.958 | *-0.00200, 0.966* | 0.07737, 0.9641 | 0.128, 0.963 |
| Labrador 2 | −0.0194, 0.914 | −0.0820, 0.931 | *-0.00244, 0.931* | 0.0506, 0.931 |
| Labrador 3 | −0.0557, 0.930 | −0.127, 0.943 | −0.0255, 0.929 | *0.00446, 0.9362* |

**Table 12.** Comparison of sPOC absolute bias and correlation of 12 single simulations run by crossing the 4 parameter sets (the default and 3 optimised sets produced by the GA at 3 locations) with 3 locations. *Italics* marks the diagonal with equal location and parameter set

| Location | Parameter Set | | | |
|----------|---------|------------|------------|------------|
| | default | Labrador 1 | Labrador 2 | Labrador 3 |
| Labrador 1 | 0.235, 0.853 | *0.00172, 0.854* | −0.00755, 0.830 | 0.0510, 0.850 |
| Labrador 2 | 0.247, 0.822 | −0.00710, 0.819 | *0.00939, 0.808* | 0.0574, 0.812 |
| Labrador 3 | 0.194, 0.850 | −0.0782, 0.854 | −0.0886, 0.813 | *-0.0325, 0.836* |

**Table 13.** Comparison of bPOC absolute bias and correlation of 12 single simulations run by crossing the 4 parameter sets (the default and 3 optimised sets produced by the GA at 3 locations) with 3 locations. *Italics* marks the diagonal with equal location and parameter set.

## 4 Discussion

A set of experiments was designed to test the potential of a newly developed GA. As a validation, the GA was first tested against
the output of a simulation produced with a known default parameter settings. The first set of experiments used a broad selection
of 9 parameters and guided our choice of those parameters that can be constrained with confidence from the evaluated variables
(in this case, sPOC and bPOC). In addition, in this set (and all others in the paper) 5 identical experiments were run at a time
and all results were similar to each other —this indicates that this method behaves consistently and reliably. The next set of
experiments was identical to the previous set, except that there were only 5 parameters selected from the initial 9-parameter set.
This particular set of experiments produced results that were closer to the default parameter with less computation. This leads





us to believe that the size of the experiment required is dependant on the size of the parameter set. The 5 parameter experiment against the default data was also compared to a similarly structured random search algorithm. Our GA produced parameter sets quicker and more consistently than the random search, providing evidence that the GA can be more computationally efficient than a random search or brute force approach. One of the main contributions of this work is to use a state-of-the-art ocean

model as a prior step to the calculation of the fitness function, with all the complexity in terms of methodology, infrastructure and resources that this option entails. This is only possible because of the aforementioned availability of computing power, and it is also highly facilitated by the usage of advanced scientific workflow solutions, that allows the integration of the model executions in the evolutionary workflow (Oana and Spataru, 2016; Dueben and Bauer, 2018; Rueda-Bayona et al., 2020).

After the experiments against default data, the GA was then tested by using observed data from ocean floats in the North

Atlantic as the reference data. A set of 5 experiments was run for each BGC-Argo float, using the same settings as in the previous set. From 10, we can see that the results show a similar level of consistency as those with the default data. There is a visible improvement in the outputs of the simulations that use a set of parameters that have been optimised by the GA compared to the outputs with the default parameter simulations (figures 13 and 14). However, most of the optimised parameter values tended rapidly towards the optimisation bounds (table 2), and exceeded them thereafter because parameters were allowed to

exceed the bounds by a small percentage each generation. This behaviour makes us question whether the optimised values are realistic, although it is also possible that we imposed too-strict bounds in some cases, given the wide range of plausible ranges that characterizes some parameters (see below). The problem of obtaining a 'right answer for the wrong reasons' is common to all PO methods when applied to complex and heavily parameterised systems (Loeptien and Dietze, 2019; Kriest et al., 2020). Therefore, PO must always be followed by a critical evaluation of the results. If a parameter converges repeatedly

to unrealistic values, regardless of the value of other parameters, this may indicate that a process is poorly represented by the model equations. In such cases, PO can prompt further model development.

Another concern that arises from the results is the need to carefully evaluate the behaviour of the cost function. This is well illustrated by figures 4 and 5, which show that on occasions some statistics were improved at the expense of others, for example bias in favour of correlation. Correctly balancing bias, dispersion and pattern (correlation) statistics in the cost

function is critical to obtain meaningful PO results. Traditional cost functions based solely on the RMSE tend to reward solutions with too-low variability, whereby the positive biases cancel out the negatives (Jolliff et al. (2009) and references therein). Cost functions as the ST score used here (Jolliff et al., 2009) were designed to avoid this problem. However, their behaviour is also sensitive to the overall variability and the signal-to-noise ratio in the data. Our preliminary tests suggested slightly better GA performance after log-transformation of the data. This procedure reduced the weight of the very high POC

concentrations present only in the surface layer in spring-summer, favouring the representation of the portions of the water column with lower POC (i.e., the mesopelagic). Unlike the model outputs used to test the GA in the first set of experiments, the BGC-Argo POC estimates are noisy. Therefore, the cost function may have been less effective when faced with the observed data. The performance of the cost function could also be improved by applying different weights to each variable, which is common practice when the reference variables exhibit very different variability ranges (Friedrichs et al., 2007; Ayata et al.,





2013). Further work quantifying the effectiveness of the cost functions across different situations should be done in order to improve the efficacy of the GA.

As a further test on our approach, two more sets of experiments were carried out in different locations in the North Atlantic, resulting in a reasonable consistency of the optimal parameter set across the region. To confirm this, the optimal parameter sets for the three locations were cross referenced by using each parameter set in each of the other two locations in single

simulations. Results from this cross-testing suggest that the parameters produced have the potential of being representative of the region or even exchangeable among some locations (figure 11), meaning that the GA is not compensating for other errors (e.g. physics) by overfitting (Loeptien and Dietze, 2019; Kriest et al., 2020). This is an important aspect because it means the GA could be used to investigate the large scale spatial variability of key biogeochemical parameters. In particular, in the past decade several authors have investigated with different approaches the latitudinal variability of the transfer efficiency of POC

from the surface ocean to its interior, arriving to contrasting conclusions (Henson et al., 2011; Marsay et al., 2015; Guidi et al., 2015; Weber et al., 2016; Schlitzer, 2004; Wilson et al., 2015). Such spatial variability would be very difficult to establish in a three-dimensional framework because of the high computational cost required. This is an example of the still open scientific questions that could be tackled with our approach.

The optimisation of PISCESv2 parameters against BGC-Argo presented in our study illustrates how PO can help understand

a dynamical system. Here we will briefly discuss the lessons learned from the **O5** experiments, while keeping in mind that a detailed review of PISCESv2 parameter values and their biogeochemical implications are beyond the scope of this paper. It is also noteworthy that the interpretation provided here draws only from the analysis of the best-performing parameter set in each GA experiment. Full exploitation of the results, with thousands of alternative model realizations, could yield further insights on how parameters interact in a space constrained by optimal model performance.

In the three **O5** experiments, *wsbio* converged to values between 0 and 1 m d$^{-1}$. The decrease in sPOC sinking speed improved the fit to observations by removing the plumes of sinking sPOC that formed below intense phytoplankton blooms in the simulations (fig. 13). Galí et al. (in prep.) identified these plumes as the main reason for sPOC PISCES-data misfit in the upper mesopelagic in several Northern and Southern hemisphere subpolar locations. Adjustment of *wsbio* effectively turned the sPOC fraction into suspended POC, which is plausible according to field studies that sorted POC fractions according to their

sinking speed (Riley et al., 2012; Baker et al., 2017). The evolution of the remaining parameters acted to adjust the magnitude and shape of POC vertical profiles. Increased *xremip* implies a steeper vertical decrease of both sPOC and bPOC, with a stronger effect on sPOC given its much longer residence time in the mesopelagic. The *xremip* parameter represents the maximal, temperature-normalized specific POC remineralization rate attainable in the model. Our optimised *xremip*, 0.078–0.114 d$^{-1}$, is consistent with the median of the highest values found across 6 field and laboratory studies when adjusted to 0 degrees:

0.10 d$^{-1}$ (Belcher et al. (2016) and references therein) with an absolute maximum of 0.135 d$^{-1}$ (Iversen and Ploug, 2010). Decreased sPOC sinking speed and increased remineralization would deplete mesopelagic sPOC if they were not compensated by other processes. In our PO experiments this deficit was compensated by increased surface microphytoplankton mortality and aggregation (*wchld*, *wchldm*), which directly supplies sPOC and bPOC. Interpretation of the evolution of the *grazflux* parameter is more complex. The flux feeding rate depends on the product of zooplankton biomass, *grazflux* and particle sinking speed.





In PISCESv2, a fraction of the intercepted bPOC is fragmented into sPOC. Therefore, flux feeding acts by removing POC (preferentially the fast-sinking bPOC) and simultaneously producing sPOC, this process becoming an important sPOC source in the lower mesopelagic (figures S1 and S2). Although a decrease in *grazflux* provided the best fit to observations, an increase in *grazflux* could also improve model skill, as shown in fig. 10 and 11. This dual behaviour gives further evidence on the difficulty of constraining this important parameter, which was highlighted in previous studies (Jackson, 1993; Stemmann et al.,

2004; Gehlen et al., 2006; Stukel et al., 2019).

To conclude this discussion, it must be highlighted that the test case chosen to evaluate the GA is an exigent one because model skill was already very good with the default parameters, despite PISCESv2 was not originally tuned to fit these particular observations. Mesopelagic POC dynamics was chosen as a test case because it is the subject of ongoing research efforts in our group and an active research front in the wider ocean sciences community (Martin et al., 2020). To illustrate the wider

applicability of our approach, we faced the GA with a different PO problem. In this additional experiment, 4 PISCES parameters that control phytoplankton growth through the utilization of nutrients and light were optimised against the seasonal cycle of sea-surface chlorophyll *a* in the Tasman Sea (figures S6-S8). As a result, the seasonal correlation between satellite-observed and PISCES-simulated chlorophyll *a* changed from $-0.45$ (default parameters) to $0.77$ (optimised parameters), accompanied by a major bias reduction. In this experiment, which started from a state of very poor model performance, the three skill metrics

that compose the ST score improved simultaneously during the optimisation. From these results we deduce that the reciprocal compensation between skill metrics, observed in the mesopelagic POC experiments, may indicate that the optimisation is operating close to the best skill attainable with a given set of model equations.

## 5 Conclusions

The GA developed shows potential in effectively constraining the parameters of the NEMO-PISCES ocean biogeochemistry

model in a way that can be extended to similar models. Importantly, the algorithm is faster and more computationally efficient than a brute-force or random approach to tune model parameters. Our GA is embedded in the workflow manager *autosubmit*, which seamlessly handles thousands of individual simulations alongside the GA calculations. This key feature makes the process of objective parameter optimisation automatic, reproducible, and portable across different high performance computing platforms.

We proposed an experimental protocol that consists of two main phases. First, the optimisation runs against the output of the default model, whose parameter values are known beforehand, to identify the parameters that can be effectively constrained when the evaluation data can be perfectly matched by the model. Second, the subset of selected parameters is optimised against the actual observations. This protocol increases the efficiency and robustness of the optimisation by reducing the parameter space.

Based on the experience acquired through the development of this tool, we make three main recommendations that can maximize the efficacy of the GA for a given research problem:





    – it may be necessary to adjust the GA metaparameters to optimise the balance between convergence speed and parameter space exploration;

– the cost function has to be selected keeping in mind the trade-offs between bias, dispersion and pattern (correlation) statistics, and a single formula is unlikely to serve all purposes equally well;

    – realistic parameter bounds are key to ensure that the results produced are sensible from a scientific point of view, and the optimisation results have to be critically evaluated a posteriori.

The use of POC estimates from BGC-Argo floats for the optimisation of biogeochemical parameters is a novel approach, as previous studies generally used target variables such as chlorophyll-*a*, nutrients or oxygen, or sparse process rate measurements
(primary production, vertical particle fluxes). The joint use of ocean observations from autonomous instruments and objective optimisation techniques is a powerful tool to improve the predictive skill of Earth System models.



## Appendix A: Example of POC sources and sinks over the annual cycle in PISCES-v2

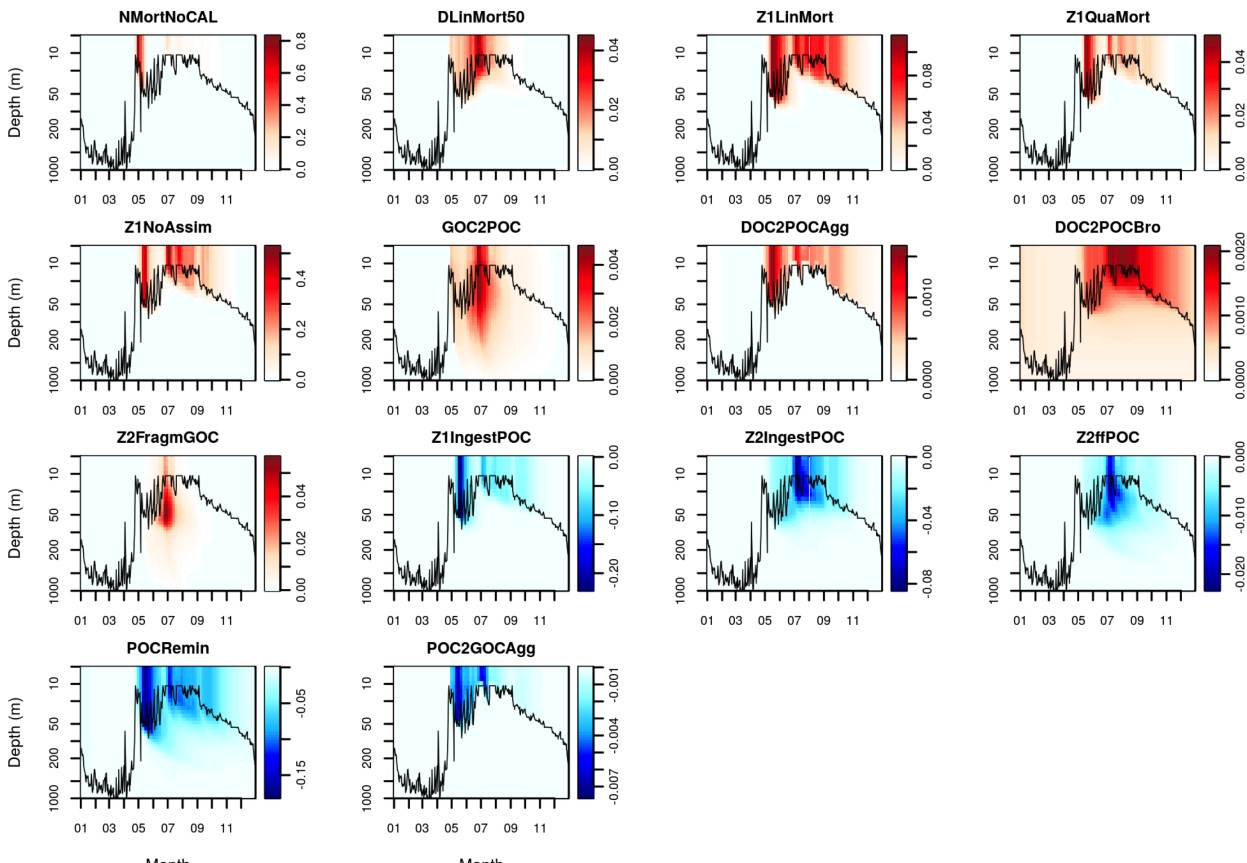

**Figure A1.** Sources (red) and sinks (blue) of the PISCES tracer POC (small detrital particulate organic carbon) over the annual cycle in the Labrador Sea with default model parameters. Rates are in mmol C m$^{-3}$ d$^{-1}$. From left to right and from to top bottom: non-calcifying nanophytoplankton mortality (NMortNoCAL); 50% diatoms mortality (DLinMort50; controlled by *wchld*); microzooplankton mortality (Z1LinMort); (Z1QuaMort); unassimilated fraction of total microzooplankton ingestion (Z1NoAssim); GOC-to-POC breakdown upon bacterial solubilization (GOC2POC; controlled by *solgoc*); DOC-to-POC aggregation caused by turbulence (DOC2POCAgg) and Brownian motion (DOC2POCBro); GOC fragmentation upon mesozooplankton flux feeding (Z2FragmGOC; controlled by *grazflux*); microzooplankton POC ingestion (Z1IngestPOC); mesozooplankton POC ingestion (Z2IngestPOC); mesozooplankton flux feeding on sinking POC (Z2ffPOC; controlled by *grazflux*); POC degradation (POCRemin; controlled by *xremip*); and POC-to-GOC aggregation caused by turbulence and differential settling (POC2GOCAgg). POC sinking rates are not shown. Details on the POC parameterisation in PISCES-v2 can be found in Aumont et al. (2015).



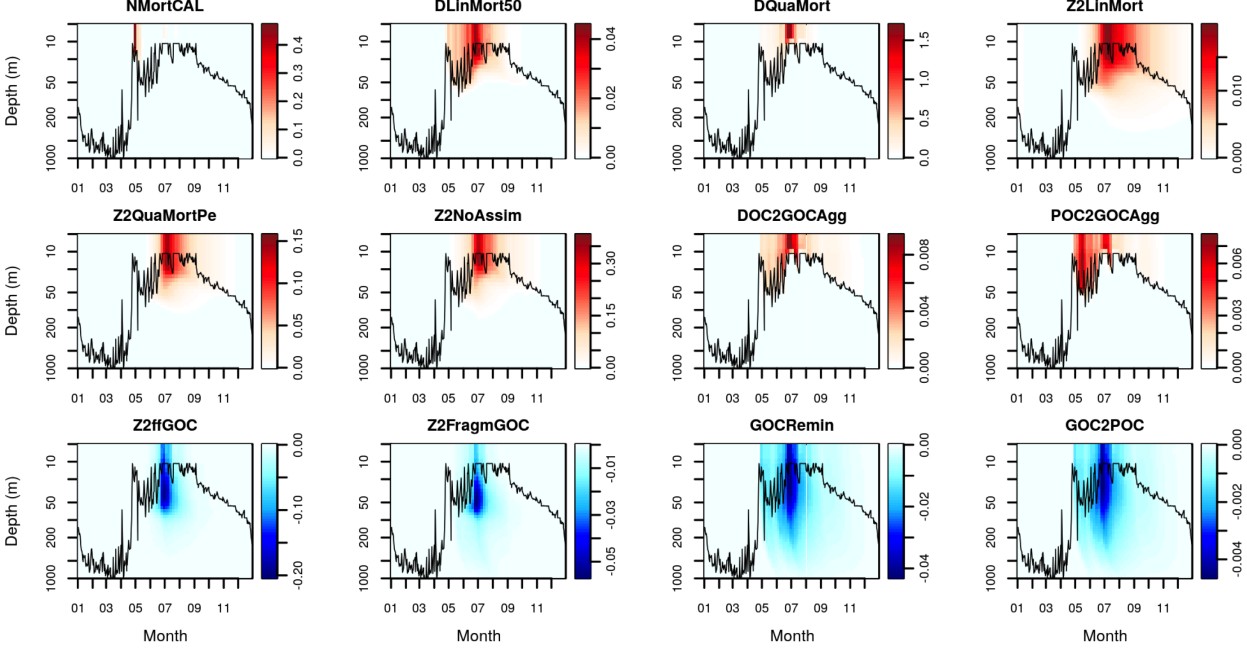

**Figure A2.** Sources (red) and sinks (blue) of the PISCES tracer GOC (large detrital particulate organic carbon) over the annual cycle in the Labrador Sea with default model parameters. Rates are in mmol C m$^{-3}$ d$^{-1}$. From left to right and from to top bottom: calcifying nanophytoplankton mortality (NMortCAL); 50% diatoms mortality (DLinMort50; controlled by *wchld*); diatoms aggregation (DQuaMort; controlled by *wchldm*); mesozooplankton mortality (Z2LinMort), fecal pellet production upon predation on mesozooplankton by upper trophic levels (Z2QuaMortPe); unassimilated fraction of total mesozooplankton ingestion (Z2NoAssim); DOC-to-GOC aggregation caused by turbulence and Brownian motion (DOC2GOCAgg); POC-to-GOC aggregation caused by turbulence and differential settling (POC2GOCAgg); mesozooplankton flux feeding on sinking GOC (Z2ffGOC; controlled by *grazflux*), GOC fragmentation upon mesozooplankton flux feeding (Z2FragmGOC; controlled by *grazflux*); GOC degradation (GOCRemin; controlled by *xremip*); and GOC-to-POC breakdown upon bacterial solubilization (GOC2POC; controlled by *solgoc*). GOC sinking rates are not shown. Details on the GOC parameterisation in PISCES-v2 can be found in Aumont et al. (2015).





## Appendix B: Sensitivity analysis for POC in PISCES-v2

**Figure B1.** Combined sensitivity of the PISCES-v2 POC tracer to the specific degradation rate of detrital organic carbon particles (*xremip*; x axis) and the sinking speed (*wsbio*, y axis). The panels show vertically-integrated POC stocks for different 4-month periods over the annual cycle (columns) and layers (rows). Epipelagic: 0-200 m; upper mesopelagic: 200-500 m; lower mesopelagic: 500-1000 m. Red crosses show the default PISCES-v2 parameter values.



**Figure B2.** Combined sensitivity of the PISCES-v2 POC tracer to the mesozooplankton flux-feeding cross section (*grazflux*; x axis) and the sinking speed (*wsbio*; y axis). The panels show vertically-integrated POC stocks for different 4-month periods over the annual cycle (columns) and layers (rows). Epipelagic: 0-200 m; upper mesopelagic: 200-500 m; lower mesopelagic: 500-1000 m. Red crosses show the default PISCES-v2 parameter values.



**Figure B3.** Combined sensitivity of the PISCES-v2 POC tracer to the specific degradation rate of detrital organic carbon particles (*xremip*; x axis) and the mesozooplankton flux-feeding cross section (*grazflux*, y axis). The panels show vertically-integrated POC stocks for different 4-month periods over the annual cycle (columns) and layers (rows). Epipelagic: 0-200 m; upper mesopelagic: 200-500 m; lower mesopelagic: 500-1000 m. Red crosses show the default PISCES-v2 parameter values.





**Appendix C: Application of the GA to optimise chlorophyll *a* in the Tasman Sea**

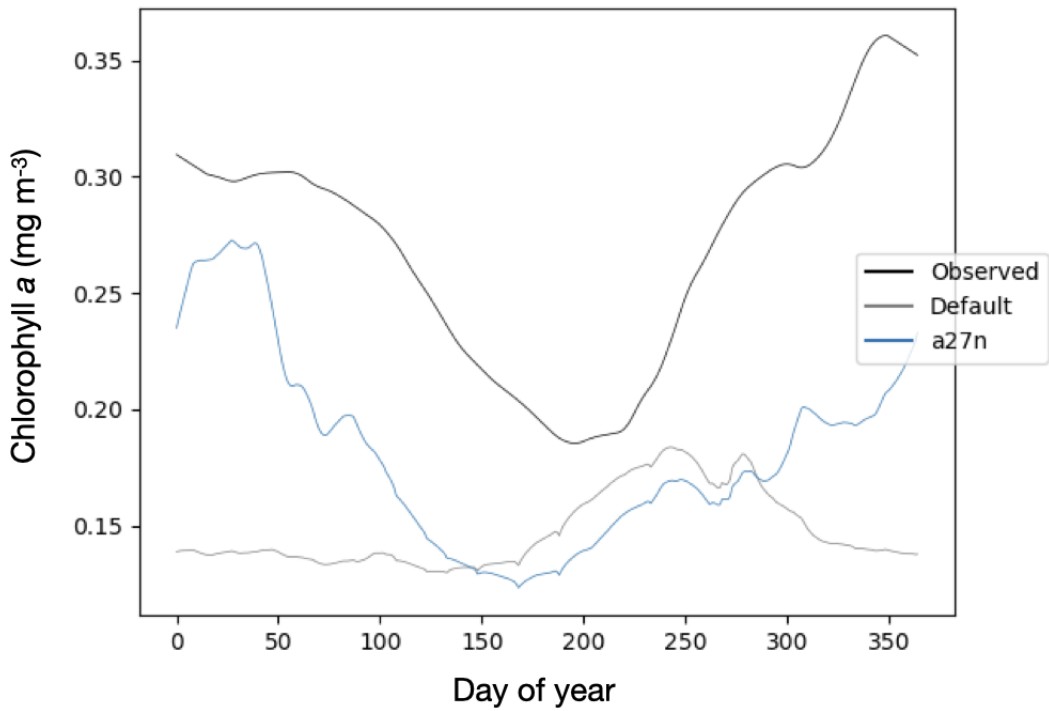

**Figure C1.** Climatological annual cycle of sea-surface chlorophyll *a* in the Tasman Sea. Satellite observations are compared to PISCES-v2 with the default parameters and with optimized parameters (GA experiment "a27n").



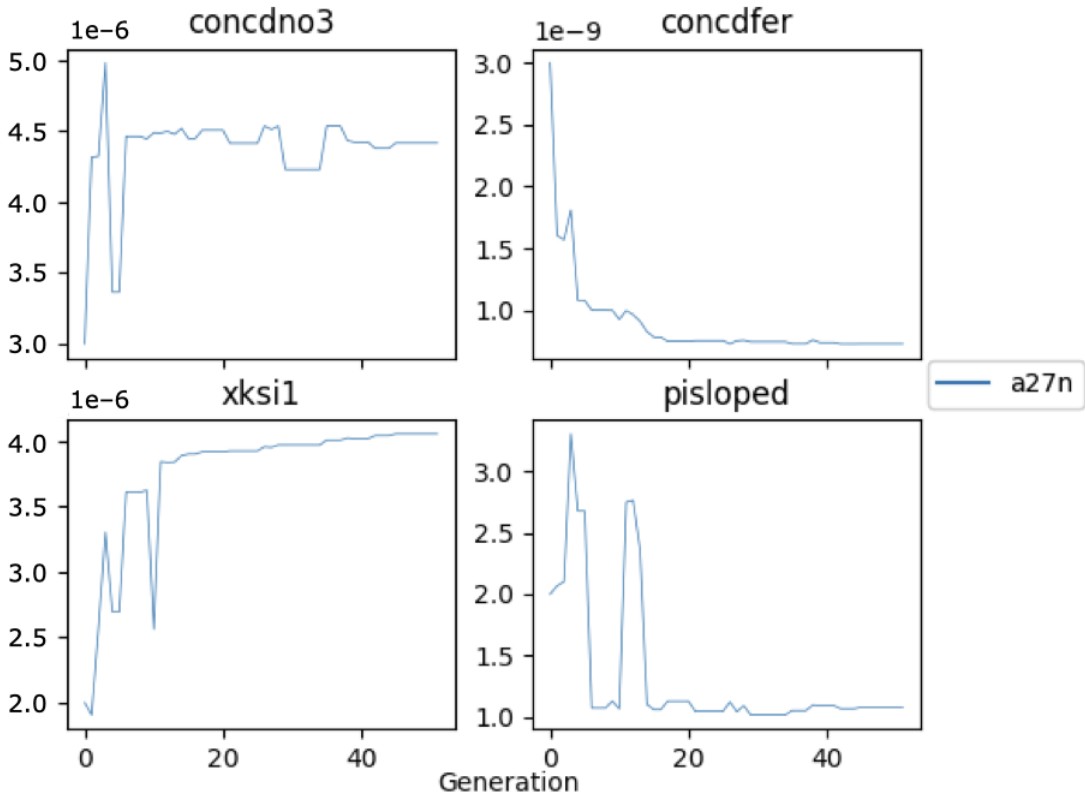

**Figure C2.** Evolution of each generation's optimal set of parameters for one single GA experiment ("a27n"; 100 individual simulations over 50 generations). The parameters included are: the nitrate half-saturation for diatoms (*concdno3*); the iron half-saturation for diatoms (*concdfer*); the half-saturation constant for silicate uptake (*xksi*); and the initial slope of the photosynthesis-irradiance curve for diatoms (*pisloped*). See Aumont et al. (2015) for details.



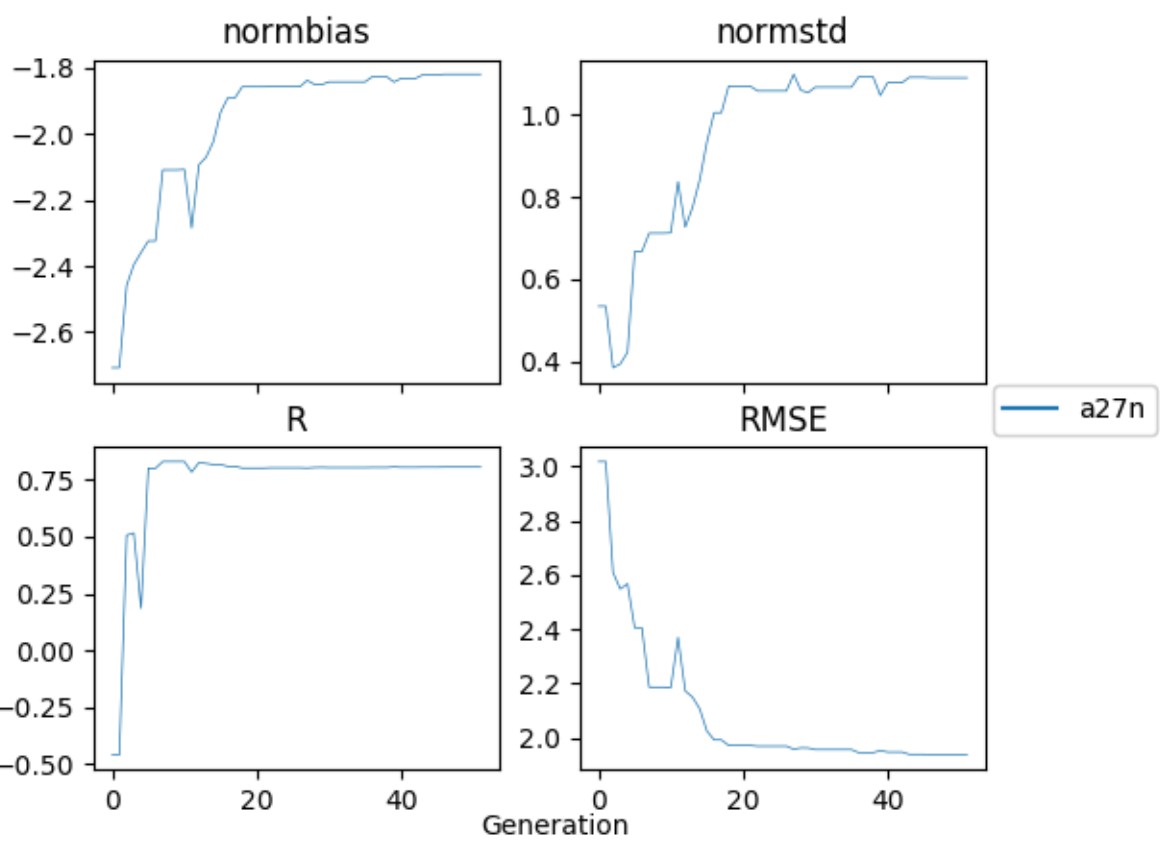

**Figure C3.** Evolution of each generation's optimal bias, normalised standard deviation, correlation and RMSE for one single GA experiment ("a27n"; 100 individual simulations over 50 generations).



*Code availability.* The code of NEMO and PISCES is publicly available at https://www.nemo-ocean.eu/. The PISCES 1D configuration used in this study is available at https://earth.bsc.es/gitlab/mgalitap/p1d_share/-/tree/gapoc/. The code for the workflow of the genetic algorithm is readily available at https://earth.bsc.es/gitlab/cp/genetic_algorithm_pisces1d

*Data availability.* These data were collected and made freely available by the International Argo Program and the national programs that contribute to it. (https://argo.ucsd.edu, https://www.ocean-ops.org). The Argo Program is part of the Global Ocean Observing System.

*Author contributions.* Marcus Falls wrote the manuscript with contributions from all co-authors. Miguel Castrillo and Mario Acosta contributed to the topics of computing and GAs. Martí Galí and Raffaele Bernardello contributed to the topics of ocean biogeochemistry. Marcus Falls developed the code of the workflow of the GA (including the implementation of the GA itself and the configuration of the workflow manager) and developed the code to produce the figures and the data in the tables. Miguel Castrillo developed the configuration of NEMO-PISCES for the MareNostrum 4 HPC platform and provided guidance to maximize the workflow efficiency. Martí Galí configured 535 the PISCES-1D offline simulations and processed the observed data from the biogeochemical Argo floats. Joan Llort provided data and guidance on biogeochemical parameters for the Tasman Sea case study. Martí Galí and Raffaele Bernardello conceived the study.

*Acknowledgements.* The simulations analysed in the paper were performed using the internal computing resources available at the Barcelona Supercomputing Center. The authors acknowledge the support of Pierre-Antoine Bretonnière and Margarida Samsó for downloading and storing the Argo data; Daniel Beltran and Wilmer Uruchi for their technical support with Autosubmit; Hervé Claustre for guidance on 540 biogeochemical Argo data processing; Olivier Aumont for guidance on PISCES-v2 structure and parameters; Xavier Yepes for guidance on the BRKGA; and Thomas Arsouze for technical support with the PISCES-1D configuration. M.G. has received financial support through the Postdoctoral Junior Leader Fellowship Programme from "La Caixa" Banking Foundation (ORCAS project; LCF/BQ/PI18/11630009) and through the OPERA project funded by the Ministerio de Ciencia, Innovación y Universidades (PID2019-107952GA-I00). R.B. acknowledges support from the Ministerio de Ciencia, Innovación y Universidades as part of the DeCUSO project (CGL2017-84493-R).

*Competing interests.* The authors declare no conflict of interest.

https://www.overleaf.com/project/5e299c28125b5500011921ef





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
