# Peer review of "Use of Genetic Algorithms for Ocean Model Parameter Optimisation: A Case Study using PISCES-v2\_RC for North Atlantic POC"

_Geoscientific Model Development, 2021_

## Referee Comment (RC1)

**Review of gmd-2021-222: *Use of Genetic Algorithms for Ocean Model Parameter Optimisation**

By Marcus Falls, Raffaele Bernardello, Miguel Castrillo, Mario Acosta, Joan Llort, and Martí Galí

**Minor Revision Recommended**

In this paper the authors describe a novel technique for tuning the parameters of an ocean biogeochemistry model using a genetic algorithm. The algorithm borrows ideas from natural selection in order to simulate the evolution of a set of parameters towards a defined optimum, which crucially depends on the definition of a cost function. The authors build a framework using AutoSubmit for conducting such an optimisation with the PISCES biogeochemical model. They try the algorithm in a few different contexts, comparing not just against a reference simulation but also directly against observations, when calculating the cost function. They find that the algorithm is much more efficient than a brute force or random search approach, though there are caveats about certain model parameters and the nature of the optimisation.

I personally found the experiments fascinating, and I am glad to see more research about genetic algorithms in the Earth Sciences, as model parameter tuning has always struck me as an obvious candidate application. The paper is well structured and the language is high-quality. It is however rather dense and difficult to read at times, so I've recommended some suggestions below. None of these suggestions require further research or experiments so I have classes this as a minor revision. I look forward to the authors' resubmission.

**Major comments**

My main criticism regards the length and density of the paper. Excluding the appendix, there are 27 figures and tables conveying a very large amount of information between them. I personally found this aspect very difficult to follow, and by page 20 I was having to switch back and forth between 3 or 4 pages with every sentence in order to corroborate the authors' statements. I would suggest to the authors to reduce the amount of information in the paper, or delegate more of it to the appendix. As a rough guide, perhaps the number of figures and tables can be reduced to 17, though I leave that to the authors' discretions. Of course, if the authors insist on keeping this amount of figures, they are welcome to. But I believe that following such a rough target will encourage the authors to digest and summarise more of the information, thereby reducing also the word count and making the reader's experience easier.

I can certainly offer some suggestions:

- Section 3.1.3: The take-home message here is that RS does better than GA. This is good news but also expected. Is it necessary to have a section just to state this?
- Does Figure 12 say anything in addition to Figure 11? It's only briefly mentioned once in the text.

- I suggest to reconsider the other tables showing numeric values at the end of the optimisations — do they really provide any more insights than the corresponding figures showing the full evolution graphically? If necessary, you can certainly mention numeric values in the text, but I'm not sure showing all of them is necessary.

**Minor comments**

- Line 82: "[Genetic algorithms] have been applied to many global search problems and, in recent years, have also started to be used in numerical modelling to avoid the limitations of today's weather and climate models." - I believe this statement but I'm not personally familiar with any examples. Could the authors provide one or more citations?
- Line 107: typo: "profiles of  the North".
- Line 199: I don't think "partition" is the natural choice of word here, and it confused me at first. Usually you "partition" something into multiple groups, but here only one group, the elites, is mentioned. I would suggest "At each generation, the $p\_e$ individuals with the best score, known as the elites, are selected, where $p\_e < p/2$".
- Line 200: Similarly, I would suggest "The remainder of the vectors are placed into the non-elite set".
- Line 201: "sets" is used twice, so now I'm confused. Is it really $p\_m$ *sets of sets* of vectors? Is it not "Next, a set of $p\_m$ randomly generated vectors are introduced…"?
- Line 204: Again, I am confused by the use of "vector sets". Is it not "A crossover in this case is a method used to generate a *new vector* by selecting two parents at random…"? The current wording implies that you re-use the same parents for all vectors produced in the crossover step, which I don't think is the case.
- Line 230: Should it be $S_3$?
- Line 318: The text refers to *wsbiomax* but the top right title in Figure 3 is *wsbio2max*. I suppose one of these is a typo.
- Figures 7 and 8: The caption mentions just D5 even though D5_rand data is also displayed.
- Line 354-355: It looks like RS is actually better than GA for "bPOC - RMSE" in Figure 9, which contradicts the statement here. The median error is lower for RS. What am I missing?
- Line 392: The optimised values for wsbio for the three experiment sets are 0.795, 0.179 and 0.008. I'm not sure why the first, O5_LAB1 is highlighted as being an outlier. All three of these numbers seem to be inconsistent with each other. Did the authors mean to highlight a different number? Perhaps they meant wchldm of O5_LAB1 which is an order of magnitude lower than the other two?
- Line 411: typo - dependent.
- Line 421: Do you mean *Figure* 10?
- I looked for supplementary figures S1 etc. but couldn't find them. I think the authors are referring to the appendix figures which have different labels. Please correct them.

---

## Referee Comment (RC2)

gmd-2021-222

Review

"Use of Genetic Algorithms for Ocean Model Parameter Optimisation"

By Marcus Falls, Raffaele Bernardello, Miguel Castrillo, Mario Acosta, Joan Llort, Martí Galí

In this paper the authors propose a Biased Random Key Genetic Algorithm (BRKGA) for the estimation of parameters of the Earth system models that ensure the optimal model performance. The method is tested using the one dimensional configuration of PISCES-v2, the biogeochemical component of NEMO, a global ocean model. In particular, a test case of particulate organic carbon is examined. First, the optimisation is done against the output of the default model. Second, the subset of selected parameters is optimised against the POC estimates from BGC-Argo floats. They find that the algorithm is faster and more computationally efficient than a brute-force or random approach to tune model parameters.

The work is very important and the approach is novel. The paper is well structured and well written. I still have some questions, answering which needs minor revision of the manuscript, I guess.

Major comments

1. In lines 322-323 you say "The results of experiment D9, plus additional analyses that we report in the Supplemental Information (SI), provided the criteria to select the 5 parameters that were used in subsequent PO experiments." I did not find Supplemental Information (SI). Therefore, I could not understand why *wsbio2* or *wsbio2max* are not selected. *wchldm* in experiment a27e converges to a much higher value than in the other experiments, thus exhibiting large spread. In the biogeochemical point of view, I can understand selection of *wchldm*. A parameter that controls sinking of bPOC is relevant and should be selected, also. In figure 14, it can be seen that bPOC has positive default and optimised bias, which could show that bPOC is not removed from the system rapidly enough. Parameter(s) that control sinking of bPOC are not optimised and the other processes do not compensate this. Please, see also comment on Figure 3 below. Therefore, I would like to see more clear justification why *wsbio2* and/or *wsbio2max* are not selected for PO.

2. My major concern is difference of optimised parameter's values in D5 and O5_LAB1 experiments. In case of O5_LAB1 experiment *wsbio* was 2-times smaller, *xremip* 3-times larger and *grazflux* almost two orders of magnitude smaller than corresponding parameters in case of D5. *Wchld* and *wchldm* had comparable values. Both of the experiments are compared at the location LAB1 (Table 3). D5 is compared to model simulation with default parameter set and O5_LAB1 with observations. Authors say that model simulation with default parameter set compared well with observations. In my understanding D5 and O5_LAB1 point to the dominant role of different processes in the same POC system, although D5 and O5_LAB1 both provide relatively good results. In D5, sPOC sinking is relatively fast and the sinks of sPOC consist of the GOC fragmentation upon mesozooplankton flux feeding (controlled by grazflux), mesozooplankton flux feeding on sinking POC (controlled by grazflux) and POC degradation (controlled by xremip). In O5_LAB1, sPOC sinking is two times slower and POC degradation (controlled by xremip) is much higher. The role of GOC fragmentation upon mesozooplankton flux feeding (controlled by grazflux) and mesozooplankton flux feeding on sinking POC (controlled by grazflux) is negligible. Concerning bPOC, the same tendency is true, except that sinking speed parameter of bPOC had default value in both cases. For instance, for me, it is difficult to decide either to use parameters' values from D5 or O5_LAB1 in the model simulation.

This rises for me a question how robust and consistent the BRKGA approach is? Is it justified to use a subset of parameters for optimisation or how the subset should be selected? Some more discussion about these issues should be provided. I acknowledge the authors discussion about the relevance of different biogeochemical processes in case of the O5 experiments.

Also, I would like the authors to mention the differences in the values of optimised parameter sets in case of O5_LAB1 in comparison to D5 in section 3.2.1.

3. L485-497 According to my opinion the inclusion of the Tasman Sea case is not natural part of this paper. Therefore, I suggest to remove this part and the figures S6-S8. If you include this part, then readers might like to have more detailed description of the experiment, etc.

Minor comments

L88: I would suggest to use term "a set of parameters" instead of "an ideal set of parameters".

L93-94: The authors claim "Finally, we discuss how our approach can become the first step towards assimilating new kinds of observations into existing Earth system models." I did not find such discussion. I suggest to remove this sentence. There is enough material in the paper even without discussion on data assimilation.

L144-145: "...here we focus on 9 parameters expected to strongly influence mesopelagic POC dynamics (table 2)." Expected by whom or why? Could you provide reference to the choice of 9 parameters or formulate it better? Is it how POC dynamics is formulated in PISCES-v2? In the next sentence you list the processes that these parameters control, and I fully agree with your choice. For readers who are not familiar with PISCES-v2, it is rather time consuming to go through mathematical formulation of the processes by Aumont et al (2015). Maybe reference to Aumont et al (2015) is sufficient.

L170: Instead "three-fold" should be "two-fold", parameter *wchldm.*

L230: I guess $S_3$ should be instead of $S_T$.

L 265: There is no reference to figure 2 in the text.

L318: Should be *wsbio2max*

Figure 3: In the experiment a27m, *wsbio2* is larger than *wsbio2max*. I seem like these two parameters "have changed their values". Also, *wchldm* in experiment a27e converges to a much higher value than in the other experiments. Could the authors have comments or discuss these cases in Discussion part? What do these cases tell about BRKGA? *caco3r* behaves differently. It shows large spread, but the end values are distributed more evenly between min and max.

L322: I did not find Supplemental Information (SI). The figures that are referred as S1, S2, etc. can be found in Appendixes, but no text on additional analyses.

L329: Should be figures A1 and A2; and B1-B3.

L341: Please specify which 3 parameters.

L357-368: Is this analysis necessary? "Both the D5 and D5_rand experiments reached the breakpoints with a similar speed, in 5–15 generations (average of 8)." "This analysis illustrates the greater efficiency of the BRKGA compared to the RS." I guess that authors mean computational efficiency. So, I would not say that the BRKGA is more efficient than the RS. No doubt that the RS outperforms the BRKGA in terms of the optimised parameter set. "Breakpoint detection can be used to stop GA

experiments in an adaptive manner, thus saving computation." Could the authors be more specific? Stopping the computation while breakpoint is reached (or after n-number of iterations) does not seem to be good idea. This paragraph, i.e. calculation of the breakpoints, does not provide additional information. Convergence of the cost function and calculated statistics can be seen in Figures and each reader could decide how many iterations are feasible and sufficient.

L371-389: I would suggest to remove the analysis related to the breakpoints. I agree that parameter values at the breakpoints are very close to their final values. But this is not the case in D5 experiments. If I would use the BRKGA, I would not stop calculations at the breakpoint or even not close to it.

L383-384: Could you provide the values of mean absolute biases (default and optimised; sPOC and bPOC) additionally to the plots in figures 13 and 14?

L412-414: See my comment L357-368. I do not see that GA produces parameter sets quicker than the RS, i.e. breakpoints were reached at the same number of iterations.

Table 9: Caption is wrong

L421: Word "figure" is missing

L451: Should reference be "table 11" instead of "figure 11"?

L467: Should be "figure"

L482-483: "…an increase in *grazflux* could also improve model skill, as shown in fig. 10 and 11." How can I see that in figures 10 and 11?

---

## Author Comment (AC2)

**General response and overview of the changes**

We thank the reviewers for the encouraging and constructive comments. To address their criticisms, we shortened the text from 512 lines to 477 lines. We have also reduced the number of display items in the main text from 27 to 18. Tables 4, 5, 6, 7, 9 and 10 have been removed because they were redundant with figures. Information on Table 1 is now given in the text. We have moved section 3.1.3, which reported on the random search (RS) algorithm, to the new Appendix C, along with Table 8 and Fig. 9 (now Table C1 and Fig. C1). We have removed the former Appendix C, which illustrated the application of the BRKGA to a different problem (optimization of the chlorophyll *a* seasonal cycle in the Tasman Sea). We have also removed the breakpoints analysis and any references to it. Finally, we have clarified or rephrased some parts of the text according to the reviewers' comments, especially to address criticisms related to the selection of parameter subsets for optimization and the difference between default and optimized parameter values depending on the type of reference data (model output vs. observations). Note also that we have changed "sPOC" to "SPOC" and "bPOC" to "LPOC" throughout for consistency with the companion paper of Galí et al. (2022, *Biogeosciences*). Detailed point by point responses to the major and minor comments are given below. Line numbers refer to the old version of the manuscript unless otherwise specified.

**Reviewer 1**

**Minor Revision Recommended**

In this paper the authors describe a novel technique for tuning the parameters of an ocean biogeochemistry model using a genetic algorithm. The algorithm borrows ideas from natural selection in order to simulate the evolution of a set of parameters towards a defined optimum, which crucially depends on the definition of a cost function. The authors build a framework using AutoSubmit for conducting such an optimisation with the PISCES biogeochemical model. They try the algorithm in a few different contexts, comparing not just against a reference simulation but also directly against observations, when calculating the cost function. They find that the algorithm is much more efficient than a brute force or random search approach, though there are caveats about certain model parameters and the nature of the optimisation. I personally found the experiments fascinating, and I am glad to see more research about genetic algorithms in the Earth Sciences, as model parameter tuning has always struck me as an obvious candidate application. The paper is well structured and the language is high-quality. It is however rather dense and difficult to read at times, so I've recommended some suggestions below. None of these suggestions require further research or experiments so I have classed this as a minor revision. I look forward to the authors' resubmission.

**Major comments**

My main criticism regards the length and density of the paper. Excluding the appendix, there are 27 figures and tables conveying a very large amount of information between them. I personally found this aspect very difficult to follow, and by page 20 I was having to switch back and forth between 3 or 4 pages with every sentence in order to corroborate the authors' statements. I would suggest to the authors to reduce the amount of information in the paper, or delegate more of it to the appendix. As a rough guide, perhaps the number of figures and tables can be reduced to 17, though I leave that to the authors' discretions. Of course, if the authors insist on keeping this amount of figures, they are welcome to. But I believe that following such a rough target will encourage the authors to digest and summarise more of the information, thereby reducing also the word count and making the reader's experience easier.

I can certainly offer some suggestions:

- Section 3.1.3: The take-home message here is that RS does better than GA. This is good news but also expected. Is it necessary to have a section just to state this?

- Does Figure 12 say anything in addition to Figure 11? It's only briefly mentioned once in the text.

- I suggest to reconsider the other tables showing numeric values at the end of the optimisations — do they really provide any more insights than the corresponding figures showing the full evolution graphically? If necessary, you can certainly mention numeric values in the text, but I'm not sure showing all of them is necessary.

**Authors**: We thank the reviewer for the encouraging and constructive comments. To address the reviewer's criticisms we have shortened the manuscript, reduced the amount of display items and removed the section on the RS, as explained in the general response.

**Minor comments**

● Line 82: "[Genetic algorithms] have been applied to many global search problems and, in recent years, have also started to be used in numerical modelling to avoid the limitations of today's weather and climate models." - I believe this statement but I'm not personally familiar with any examples. Could the authors provide one or more citations?

We have shortened the sentence and added citations.

● Line 107: typo: "profiles of in the North".

Fixed, thanks.

● Line 199: I don't think "partition" is the natural choice of word here, and it confused me at first. Usually you "partition" something into multiple groups, but here only one group, the elites, is mentioned. I would suggest "At each generation, the p_e individuals with the best score, known as the elites, are selected, where p_e < p/2".

Rephrased as suggested. We additionally replaced "elites" by "elite subpopulation" for clarity and consistency with the following sentence.

● Line 200: Similarly, I would suggest "The remainder of the vectors are placed into the non-elite set".

Replaced "non-elite set" by "non-elite subpopulation".

● Line 201: "sets" is used twice, so now I'm confused. Is it really p_m sets of sets of vectors? Is it not "Next, a set of p_m randomly generated vectors are introduced…"?

Thanks for calling our attention to this mistake. As explained in line 196, "sets of parameters" are equivalent to "vectors (of parameters)". We adopted the rephrasing proposed by the reviewer.

● Line 204: Again, I am confused by the use of "vector sets". Is it not "A crossover in this case is a method used to generate a new vector by selecting two parents at random…"? The current wording implies that you re-use the same parents for all vectors produced in the crossover step, which I don't think is the case.

Thanks. We deleted "set" as the reviewer suggested.

The sentence now reads: "A crossover in this case is a method used to generate a new vector by selecting two parents at random and then each element of the new vector is randomly picked from one of the two parents."

● Line 230: Should it be S3?

Fixed, thanks.

● Line 318: The text refers to wsbiomax but the top right title in Figure 3 is wsbio2max. I suppose one of these is a typo.

Fixed, thanks.

● Figures 7 and 8: The caption mentions just D5 even though D5_rand data is also displayed.

The figures have been modified and do not show D5_rand experiments.

● Line 354-355: It looks like RS is actually better than GA for "bPOC - RMSE" in Figure 9, which contradicts the statement here. The median error is lower for RS. What am I missing?

We have moved the section on the RS algorithm to Appendix C.

● Line 392: The optimised values for wsbio for the three experiment sets are 0.795, 0.179 and 0.008. I'm not sure why the first, O5_LAB1 is highlighted as being an outlier. All three of these numbers seem to be inconsistent with each other. Did the authors mean to highlight a different number? Perhaps they meant wchldm of O5_LAB1 which is an order of magnitude lower than the other two?

● Line 411: typo - dependant dependent.

Fixed.

● Line 421: Do you mean Figure 10?

Thanks, fixed.

● I looked for supplementary figures S1 etc. but couldn't find them. I think the authors are referring to the appendix figures which have different labels. Please correct them.

We changed the context of the Appendices and now they are correctly referenced in the text, as explained in the general response.

**Reviewer 2**

In this paper the authors propose a Biased Random Key Genetic Algorithm (BRKGA) for the estimation of parameters of the Earth system models that ensure the optimal model performance. The method is tested using the one dimensional configuration of PISCES-v2, the biogeochemical component of NEMO, a global ocean model. In particular, a test case of particulate organic carbon is examined. First, the optimisation is done against the output of the default model. Second, the subset of selected parameters is optimised against the POC estimates from BGC-Argo floats. They find that the algorithm is faster and more computationally efficient than a brute-force or random approach to tune model parameters. The work is very important and the approach is novel. The paper is well structured and well written. I still have some questions, answering which needs minor revision of the manuscript, I guess.

**Major comments**

1. In lines 322-323 you say "The results of experiment D9, plus additional analyses that we report in the Supplemental Information (SI), provided the criteria to select the 5 parameters that were used in subsequent PO experiments." I did not find Supplemental Information (SI). Therefore, I could not understand why wsbio2 or wsbio2max are not selected. wchldm in experiment a27e converges to a much higher value than in the other experiments, thus exhibiting large spread. In the biogeochemical point of view, I can understand selection of wchldm. A parameter that controls sinking of bPOC is relevant and should be selected, also. In figure 14, it can be seen that bPOC has positive default and optimised bias, which could show that bPOC is not removed from the system rapidly enough. Parameter(s) that control sinking of bPOC are not optimised and the other processes do not compensate this. Please, see also comment on Figure 3 below. Therefore, I would like to see more clear justification why wsbio2 and/or wsbio2max are not selected for PO.

We thank the reviewer for the positive evaluation of our manuscript and for the constructive criticisms. As explained in the general response, we have resolved the confusion between SI and Appendices.

The second and third paragraphs of section 3.1.1 (Nine parameters (D9)) have been slightly rephrased to better justify the selection of parameters, and now reads as follows:

"The results of experiment **D9**, plus additional analyses that we report in Appendix A, provided the

criteria to select the 5 parameters that were used in subsequent PO experiments. Quite obviously, the POC sinking speed, *wsbio*, and the specific remineralization rate of both *POC* and *GOC*, *xremip*, were selected owing to their rapid and robust convergence to the expected values. In addition, *wchld*, *wchldm* and *grazflux*, which showed vacillating convergence behaviour in **D9**, were selected owing to their important role in POC budgets in the study area. In particular, *grazflux* can greatly attenuate the gravitational LPOC flux in the upper mesopelagic while fragmenting a fraction of LPOC to SPOC. The parameters *wchld* and *wchldm* control detrital POC formation through phytoplankton mortality and aggregation, especially during phytoplankton blooms collapse. Hence, their inclusion is further justified by the need to optimize parameters that control *POC* and *GOC* sources, not only sinks (controlled by the other parameters).

Figures A1 and A2 show the contribution of individual source and sink terms to the *POC* and *GOC* rates of change with the default parameter set, demonstrating the important role of the 5 selected parameters. Additional experiments (not shown) were run with *solgoc* and other parameters not included in table 1, supporting the choice of the previous 5 parameters. Still, we acknowledge that the selection of a subset of parameters is somewhat arbitrary and other parameters that showed some potential for being constrained might be included in future experiments".

In the Discussion, we have added the following paragraph to acknowledge the limitations of our study:

"The selection of a subset of model parameters is a common limitation of PO experiments and, although we based it on objective criteria, we acknowledge it remains somewhat arbitrary. The stepwise reduction of the number of parameters from 9 to 5 obeys the need to assess the GA performance with a varying number of parameters, and also to reduce the degrees of freedom given that only 2 variables were used as reference observations. Among the excluded parameters, *wsbio2* certainly deserves examination in future experiments given its primary control on the fate of mesopelagic *GOC* (large detritus). There are three main reasons that led us to exclude *wsbio2* from this work: (1) our optimisation exercise focused on POC stocks, which are largely dominated by the SPOC fraction that typically represents around 85% of total POC (Galí et al. 2022, and references therein); (2) experiment set **D9** suggested that, unlike *wsbio*, *wsbio2* might be difficult to constrain even from error-free observations provided by the default model run; and (3) estimation of *wsbio2* relies more than any other parameter on LPOC observations, which suffer from large uncertainty in comparison to SPOC (Galí et al., 2022); ."

The large uncertainty in LPOC was also highlighted in the initial manuscript, line 388: "Further reduction in the bPOC [now LPOC] misfit could have been impeded by the noisier nature of the observed bPOC data". Finally, it must be noted that the main objective of the current paper is to introduce the BRKGA and its workflow and to show that mesopelagic POC as seen by BGC-Argo floats provides a viable case for optimisation.

2. My major concern is the difference of optimised parameter's values in D5 and O5_LAB1 experiments. In case of O5_LAB1 experiment wsbio was 2-times smaller, xremip 3-times larger and grazflux almost two orders of magnitude smaller than corresponding parameters in case of D5. Wchld and wchldm had comparable values. Both of the experiments are compared at the location LAB1 (Table 3). D5 is compared to model simulation with default parameter set and O5_LAB1 with observations. Authors say that model simulation with default parameter set compared well with observations. In my understanding D5 and O5_LAB1 point to the dominant role of different processes in the same POC system, although D5 and O5_LAB1 both provide relatively good results. In D5, sPOC sinking is relatively fast and the sinks of sPOC consist of the GOC fragmentation upon mesozooplankton flux feeding (controlled by grazflux), mesozooplankton flux feeding on sinking POC (controlled by grazflux) and POC degradation (controlled by xremip). In O5_LAB1, sPOC sinking is two times slower and POC degradation (controlled by xremip) is much higher. The role of GOC fragmentation upon mesozooplankton flux feeding (controlled by grazflux) and mesozooplankton flux feeding on sinking POC (controlled by grazflux) is negligible. Concerning bPOC, the same

tendency is true, except that sinking speed parameter of bPOC had default value in both cases. For instance, for me, it is difficult to decide either to use parameters' values from D5 or O5_LAB1 in the model simulation. This rises for me a question how robust and consistent the BRKGA approach is? Is it justified to use a subset of parameters for optimisation or how the subset should be selected? Some more discussion about these issues should be provided. I acknowledge the authors discussion about the relevance of different biogeochemical processes in case of the O5 experiments. Also, I would like the authors to mention the differences in the values of optimised parameter sets in case of O5_LAB1 in comparison to D5 in section 3.2.1.

In response to the last sentence, we have modified the first sentences of section 3.2.1 to highlight the different evolution of the parameters in O5_LAB1 compared to D5: "Figure 9 shows the evolution of the optimal set of parameters in each generation of experiment set O5_LAB1. Two types of behaviour are observed: the parameters wchld and wchldm converge to a range that brackets the default values, whereas wsbio, grazflux and xremip clearly deviate from the default values. Still, the latter three parameters behave consistently across the 5 replicate experiments, which is in line with how they behave in D5".

Regarding the reviewer's concerns about the differences between D5 and O5 sets of experiments, and the applicability of the O5 results, we would like to note that "a detailed review of PISCESv2 parameter values and their biogeochemical implications are beyond the scope of this paper", quoting from lines 460-461. This was also indicated in the abstract: "Given the deviation of the optimal set of parameters from the default, further analyses using observed data in other locations are recommended to establish the validity of the results obtained". Similar messages were already given in the Discussion (lines 425-431) and Conclusions (lines 516-517). Our manuscript also included a brief discussion (lines 459-485) of the biogeochemical implications of the optimized parameters, as the reviewer acknowledged.

The deviation of the optimized parameters from the default makes R2 question 'how robust and consistent the BRKGA approach is'. In our view, this deviation does not inform about the robustness of the BRKGA, but about (1) uncertainty in model parameters and equations, and (2) the need for additional types of observations. Hence, we included these sentences in the last paragraph of the Discussion: "Our approach has the additional value of showing that similarly good fits can be obtained with different parameter sets, pointing to the many degrees of freedom in the model, and calls for a continuous reassessment of model uncertainty as new observations become available. Future work is granted to study the sensitivities, interdependencies and optimal values of PISCES parameters through more comprehensive experiments".

In summary, the main goals of this paper were to demonstrate that (1) the BRKGA workflow presented is an effective, consistent and replicable approach to PO, and (2) that our system under scrutiny (mesopelagic POC) provides a workable case. In our view these goals were accomplished.

3. L485-497 According to my opinion the inclusion of the Tasman Sea case is not natural part of this paper. Therefore, I suggest to remove this part and the figures S6-S8. If you include this part, then readers might like to have more detailed description of the experiment, etc.

We agree with the reviewer. We have removed the Appendix reporting on the Tasman Sea experiment. We left one sentence in the Discussion referring to the Tasman Sea experiments because we believe they add a wider perspective on the performance of the BRKGA, line 394 of the new version:

"Ongoing work with a different optimization case indicates that the BRKGA can produce larger and simultaneous improvements in all skill metrics when starting from a state of very poor model performance, in this case the seasonal cycle of sea-surface chlorophyll *a* in the Tasman Sea (J. Llort, personal communication; data not shown). Therefore, the trade-offs between skill metrics observed here during the evolution of the experiments may indicate that the optimisation was operating close to the best skill attainable with a given set of model equations and considering observational

uncertainty".

**Minor comments**

● L88: I would suggest to use term "a set of parameters" instead of "an ideal set of parameters".

"Ideal" removed.

● L93-94: The authors claim "Finally, we discuss how our approach can become the first step towards assimilating new kinds of observations into existing Earth system models." I did not find such discussion. I suggest to remove this sentence. There is enough material in the paper even without discussion on data assimilation.

Sentence removed.

● L144-145: "…here we focus on 9 parameters expected to strongly influence mesopelagic POC dynamics (table 2)." Expected by whom or why? Could you provide reference to the choice of 9 parameters or formulate it better? Is it how POC dynamics is formulated in PISCES-v2? In the next sentence you list the processes that these parameters control, and I fully agree with your choice. For readers who are not familiar with PISCES-v2, it is rather time consuming to go through mathematical formulation of the processes by Aumont et al (2015). Maybe reference to Aumont et al (2015) is sufficient.

We rephrased as follows: "9 parameters (table 2) expected to strongly influence mesopelagic POC dynamics according to model equations (Aumont et al. 2015, 2017) and preliminary analyses (Appendices A and B)"

● L170: Instead "three-fold" should be "two-fold", parameter wchldm.

Thanks, fixed.

● L230: I guess S3 should be instead of ST.

Thanks, fixed.

● L 265: There is no reference to figure 2 in the text.

Reference added in Section 2.4.

● L318: Should be wsbio2max.

Thanks, fixed.

● Figure 3: In the experiment a27m, wsbio2 is larger than wsbio2max. I seem like these two parameters "have changed their values". Also, wchldm in experiment a27e converges to a much higher value than in the other experiments. Could the authors have comments or discuss these cases in the Discussion part? What do these cases tell about BRKGA? caco3r behaves differently. It shows a large spread, but the end values are distributed more evenly between min and max.

We thank the reviewer for these appreciations. Given that we have justified the exclusion of wsbio2 and wsbio2max in the Results and the Discussion (see response to major comments above), we prefer not to expand on the analysis of these interesting behaviors to keep the article concise.

● L322: I did not find Supplemental Information (SI). The figures that are referred as S1, S2, etc. can be found in Appendices, but no text on additional analyses.

We corrected the reference to Appendices A and B, and rewrote part of the paragraph to emphasize that the Appendix figures support the selection of parameters.

● L329: Should be figures A1 and A2; and B1-B3.

Thanks, fixed.

● L341: Please specify which 3 parameters.

Fixed.

● L357-368: Is this analysis necessary? "Both the D5 and D5_rand experiments reached the breakpoints with a similar speed, in 5–15 generations (average of 8)." "This analysis illustrates the greater efficiency of the BRKGA compared to the RS." I guess that authors mean computational efficiency. So, I would not say that the BRKGA is more efficient than the RS. No doubt that the RS outperforms the BRKGA in terms of the optimised parameter set.

"Breakpoint detection can be used to stop GA experiments in an adaptive manner, thus saving computation." Could the authors be more specific? Stopping the computation while breakpoint is reached (or after n-number of iterations) does not seem to be a good idea. This paragraph, i.e. calculation of the breakpoints, does not provide additional information. Convergence of the cost function and calculated statistics can be seen in Figures and each reader could decide how many iterations are feasible and sufficient.

We removed the section devoted to RS, which includes the suggestion of using breakpoint detection to stop experiments.

● L371-389: I would suggest to remove the analysis related to the breakpoints. I agree that parameter values at the breakpoints are very close to their final values. But this is not the case in D5 experiments. If I would use the BRKGA, I would not stop calculations at the breakpoint or even not close to it.

We reduced the analysis related to breakpoints and presented it only as an indicator of the convergence speed, which also lets readers/users evaluate the benefits of extending the optimization experiments in proportion to the improvement in the fit.

● L383-384: Could you provide the values of mean absolute biases (default and optimised; sPOC and bPOC) additionally to the plots in figures 13 and 14?

We declined this suggestion because these statistics are already shown in figure 11 (formerly 12). We added a mention to figure 11 in the caption of figures 12 and 13 (formerly 13 and 14).

● L412-414: See my comment L357-368. I do not see that GA produces parameter sets quicker than the RS, i.e. breakpoints were reached at the same number of iterations.

The breakpoints analysis was removed.

● Table 9: Caption is wrong

The table has been removed.

● L421: Word "figure" is missing

Thanks, fixed (now Fig. 9)

● L451: Should reference be "table 11" instead of "figure 11"?

Thanks, fixed (now Table 3)

● L467: Should be "figure"

Thanks, now figure 12

● L482-483: "…an increase in grazflux could also improve model skill, as shown in fig. 10 and 11." How can I see that in figures 10 and 11?

Rephrased as follows: "as shown by the relatively good skill of experiment a3nj during the first few generations (Fig. 9 and 10). This dual behaviour is confirmed by the sensitivity analysis shown in Fig. B3, which shows that, for a given *xremip*, the decrease in *POC* needed to improve model-data fit in the mesopelagic can be achieved by either increasing or decreasing *grazflux*"